# A Review on Five and Six-Membered Heterocyclic Compounds Targeting the Penicillin-Binding Protein 2 (PBP2A) of Methicillin-Resistant *Staphylococcus aureus* (MRSA)

**DOI:** 10.3390/molecules28207008

**Published:** 2023-10-10

**Authors:** Shraddha S. Ambade, Vivek Kumar Gupta, Ritesh P. Bhole, Pramod B. Khedekar, Rupesh V. Chikhale

**Affiliations:** 1Department of Pharmaceutical Sciences, Rashtrasant Tukadoji Maharaj Nagpur University, Nagpur 440033, MH, Indiapbkhedekarudps@gmail.com (P.B.K.); 2Department of Biochemistry, National JALMA Institute for Leprosy & Other Mycobacterial Diseases (ICMR), Agra 282004, UP, India; 3Dr. D. Y. Patil Institute of Pharmaceutical Sciences and Research, Pimpri, Pune 411018, MH, India; 4Dr. D. Y. Patil Dental College and Hospital, Dr. D. Y. Patil Vidyapeeth, Pimpri, Pune 411018, MH, India; 5UCL School of Pharmacy, 29-39 Brunswick Square, London WC1N 1AX, UK

**Keywords:** penicillin-binding protein 2a (PBP2a), mur enzymes, heterocycles, high-throughput screenings, transpeptidation

## Abstract

*Staphylococcus aureus* is a common human pathogen. Methicillin-resistant *Staphylococcus aureus* (MRSA) infections pose significant and challenging therapeutic difficulties. MRSA often acquires the non-native gene PBP2a, which results in reduced susceptibility to β-lactam antibiotics, thus conferring resistance. PBP2a has a lower affinity for methicillin, allowing bacteria to maintain peptidoglycan biosynthesis, a core component of the bacterial cell wall. Consequently, even in the presence of methicillin or other antibiotics, bacteria can develop resistance. Due to genes responsible for resistance, *S. aureus* becomes MRSA. The fundamental premise of this resistance mechanism is well-understood. Given the therapeutic concerns posed by resistant microorganisms, there is a legitimate demand for novel antibiotics. This review primarily focuses on PBP2a scaffolds and the various screening approaches used to identify PBP2a inhibitors. The following classes of compounds and their biological activities are discussed: Penicillin, Cephalosporins, Pyrazole-Benzimidazole-based derivatives, Oxadiazole-containing derivatives, non-β-lactam allosteric inhibitors, 4-(3*H*)-Quinazolinones, Pyrrolylated chalcone, Bis-2-Oxoazetidinyl macrocycles (β-lactam antibiotics with 1,3-Bridges), Macrocycle-embedded β-lactams as novel inhibitors, Pyridine-Coupled Pyrimidinones, novel Naphthalimide corbelled aminothiazoximes, non-covalent inhibitors, Investigational-β-lactam antibiotics, Carbapenem, novel Benzoxazole derivatives, Pyrazolylpyridine analogues, and other miscellaneous classes of scaffolds for PBP2a. Additionally, we discuss the penicillin-binding protein, a crucial target in the MRSA cell wall. Various aspects of PBP2a, bacterial cell walls, peptidoglycans, different crystal structures of PBP2a, synthetic routes for PBP2a inhibitors, and future perspectives on MRSA inhibitors are also explored.

## 1. Introduction

Methicillin-resistant *Staphylococcus aureus* (MRSA) is often referred to as a superbug [1]. When we hear the term MRSA, we immediately think of *S. aureus*. To better understand MRSA, we need to learn about *S. aureus*, its disease-causing capacity, and its resistance mechanisms. MRSA is produced when methicillin-susceptible *S. aureus* (MSSA) acquires the mecA gene, which is a mutated version. This means that methicillin resistance depends on the presence of the mecA gene [2]. *S. aureus* is a common human pathogen as well as a commensal bacterium. It causes a variety of clinical infections, including bacteremia, some device-related infections (such as implantable cardiac device infections, intravascular catheter infections, and other prosthetic device-related infections), pleuropulmonary infections, osteoarticular infections (osteomyelitis, native joint septic arthritis, and prosthetic joint infections), skin and soft tissue infections, and infective endocarditis. However, it rarely causes urinary tract infections, although there have been reported cases [3]. Osteomyelitis, Ritter’s disease, endocarditis, and bacteremia are also caused by *S. aureus* in the developed world [4], and septicemia can also be attributed to *S. aureus* [5].

Commonly, *S. aureus* is responsible for infections both in the community and among healthcare professionals. These infections can vary from superficial skin and soft tissue infections to invasive infections. They can also lead to sepsis and death. This pathogen can adapt to various environmental conditions, making *S. aureus* a major concern [6,7]. Due to its wide range of infection-causing abilities, *S. aureus* is often referred to as a “superbug” [8]. *S. aureus* (MRSA) appears to be the most frequently diagnosed bacteria in many parts of the world. This pathogen is resistant to all β-lactam antibiotics and various other classes of antibiotics that can inhibit *Staphylococcus aureus* [9]. Various antibiotics, such as β-lactam antibiotics, are used to treat these infections. The ability of β-lactam antibiotics to access and interact with penicillin-binding proteins is crucial for their effectiveness against bacteria [10]. In the Staphylococcus family, twenty-nine species and eight subspecies have been identified. Due to the diversity of species, understanding and identifying the reasons for resistance can be a challenging task [11]. Millions of deaths occur each year due to antibiotic resistance, leading to various forms of infection [12]. *S. aureus* is responsible for nosocomial and community-acquired bacterial infections worldwide [13].

To combat penicillin-resistant *Staphylococcus aureus* [14], methicillin, a penicillinase-resistant penicillin, was introduced in 1959. However, within a year, Professor Patricia Jevons reported the first methicillin-resistant human *S. aureus* strains in a hospital in the UK [15]. In the early 1960s, the introduction of semisynthetic β-lactam antibiotics like methicillin and oxacillin led to a general decrease in the frequency of multidrug-resistant bacteria [16]. As bacteria develop resistance to different antibiotics, various forms of infections, such as VRSA, MSSA, and MRSA, can result. MRSA has been found to be the most common gram-positive pathogen in nosocomial infections. According to the National Nosocomial Infections Surveillance (NNIS), it is mainly associated with infections in intensive care units [17]. The first major nosocomial epidemic of MRSA was described in 1963. In the 1970s, the USA reported nosocomial infections caused by MRSA. Methicillin resistance can be of two types: homogeneous resistance and heterogeneous resistance. Heterogeneous strains may consist of two populations, one relatively susceptible and the other highly resistant. In contrast to homogeneous strains, which contain a single cell population, all these strains tend to be highly resistant to *S. aureus*. Some strains may even produce toxins that can trigger primary and secondary bacteremia and various infections [18,19]. When methicillin-resistant *S. aureus* was first described to the medical community, it was a significant publication, published in a British medical journal fifty years ago [20]. Outside of the healthcare setting, MRSA has become a common pathogen. The first cases of MRSA in the community were reported in the late 1980s, affecting patients who had not been exposed to hospitals or nursing homes [21].

### History and Emergence of MRSA

MRSA is a multi-drug-resistant bacterium that causes serious infections in both communities and hospitalized patients worldwide, along with diseases like tuberculosis as reported in [22]. Methicillin-resistant *Staphylococcus aureus* (MRSA) is responsible for outbreaks of nosocomial infections and community-associated infections reported worldwide [23]. Multilocus sequencing (MLST) has identified five different clonal complexes: CC45, CC30, CC22, CC8, and CC5, which constitute 88.2% of all MRSA infections [24,25]. The first MRSA strains resistant to various antibiotics emerged in the second half of the 1960s, and multidrug-resistant MRSA has been identified in several central European countries [26], including England, as well as Australia and India. MRSA strains with simultaneous susceptibility to penicillin, streptomycin, and tetracycline, but rarely erythromycin, accounted for 15% of all *S. aureus* strains reported in Denmark in 1971 [27]. Infections acquired in intensive care units account for nearly 60% of nosocomial *S. aureus* infections in the USA [28]. MRSA strains have been responsible for nosocomial *S. aureus* surgical site and bloodstream infections in the USA, UK, Japan, Finland, Turkey, the Netherlands, and many other countries [29]. These include the Americas, Europe, the Middle East, North Africa, and East Asia, where MRSA has become a significant pathogen in both nosocomial and community-based settings over the last decade [30].

MRSA is considered a major contributor to multidrug-resistant nosocomial infections, including bloodstream infections and nosocomial pneumonia. *S. aureus* is a gram-positive bacterium responsible for skin and respiratory infections [31]. In India, infections caused by *S. aureus* are on the rise, putting a growing strain on healthcare resources. MRSA infections in hospitals are particularly widespread [32]. Patients in ICUs, especially those with surgical wounds, drains, and invasive monitoring systems, are at increased risk of infection. Methicillin-resistant *S. aureus* has been detected in 13–47% of cases in India. Due to increasing resistance to various antibiotics, the options for treating MRSA are diminishing [33]. An outbreak of MRSA infection was reported in a teaching hospital located in North-West India [34].

Methicillin-resistant *S. aureus* is one of the most common bacteria, causing complicated skin and skin structure infections (cSSSIs) and other hospital-acquired illnesses, such as bloodstream infections (BSIs) and ventilator-associated pneumonia (VAP) [35]. According to data from the Animal Quarantine Service of the Japanese Ministry of Agriculture, Forestry, and Fisheries, MRSA ST398 has been transmitted through the swine production system. Surveys were conducted in Asian countries like China, Malaysia, and Singapore to assess the prevalence of MRSA in Japanese pigs. It appears that isolated MRSA strains remain resistant to antibiotics. Even with the use of an enrichment culture method with a 7.5 percent NaCl broth medium, the isolation rate of MRSA from pigs in Japan (0.9 percent) appears to be lower than in Europe and North America [36] (for more information, refer to http://www.maff.go.jp/aqs/tokei/toukeinen.html, accessed on 15 May 2013). In 2011, an MRSA outbreak was noted in the special care baby unit in the United Kingdom at Cambridge University Hospital National Health Service Foundation Trust (CUH) [37]. Over the past decade, a significant increase in MRSA transmission and disease among healthy individuals has been documented in Taiwan. MRSA infections accounted for more than half of all pediatric community-acquired infections in up to 9.5 percent of normal Taiwanese children’s nasal passages. The total global population has also been affected by MRSA, with more than 80,000 invasive infections reported in 2011 in the United States alone and other countries such as Japan and many more, bringing the total number of people infected with MRSA globally to 53 million [38,39]. Nosocomial infections have spread in various countries, including the Netherlands, the entire Scandinavian region, Greece, France, Europe, Italy, Spain, and Germany [40]. MRSA prevalence has been observed in various other countries, including Australia (1965), the United States (1968), the UK (1970s), Japan (2003), and China (2011). Outbreaks of MRSA have been reported in these countries [41]. Due to its resistance to many antibiotics, sepsis-induced MRSA infection has a worse outcome and can lead to death from septic shock [42]. In the United Kingdom (England and Wales), efforts to prevent healthcare-associated infections have focused on actions such as improved antibiotic selection, isolation of infected patients, the use of gloves when handling them, enhanced hand hygiene, etc. These efforts led to a 50% reduction in MRSA infections from April 2010 to March 2011 throughout the National Health Service compared to the reported cases in 2008 and 2009. In 2012, there was a suspected MRSA outbreak involving 12 patients in the Neurosurgery Department at Jiangsu University, affiliated with the hospital [43,44]. It is still a subject of debate whether MRSA is more virulent than MSSA [43]. According to research and reported data, MRSA has also surpassed MSSA as the most common *S. aureus* strain in community-acquired infections. Incision and drainage with antibiotic treatment were used in 60% of cases, while incision and evacuation alone were used in 19%. In 64 percent of cases where antibiotic therapy was administered, a beta-lactam antibiotic was employed, with 57 percent of MRSA patients receiving antibiotic treatment (Figure 1). This approach resolved 96.1% of cases, with no association with antimicrobial medication. The need for drainage of a cutaneous abscess is universally recognized, but the reasons for using antibiotics in conjunction with drainage remain unclear [44]. CA-MRSA infections differ in many aspects from HA-MRSA infections. The epidemiological differences between different strains of the same clonal CA-MRSA complex are also unclear. The first case of CA-MRSA was reported in 1993 in Western Australia, and MRSA infection was initially identified in Australia in 1965. The first case of hospital MRSA was reported in Boston in 1968 [41].

## 2. PBPs, Types of PBP, Their Location, and Contribution in Transpeptidation Reaction

Penicillin-binding proteins (PBPs) are the target of β-lactam antibiotics due to their propensity to bind radio-labeled penicillin, making them easily detectable. PBPs are serine acyltransferases that facilitate the production of cross-linked peptidoglycan and serve as targets for β-lactam antibiotics due to their transpeptidase-related catalytic activity. β-lactam antibiotics covalently bind to the active site serine of PBPs after adhering to the PBP catalytic cleavage, resulting in the formation of a slowly hydrolyzed acyl-enzyme complex that reduces peptidoglycan cross-linking [46,47]. PBPs are present in the periplasmic membrane of bacterial cell walls and are membrane-bound transpeptidases that allow antibiotics to reach the active site without passing through the cytoplasmic membrane [48,49].

PBPs are historically divided into high molecular weight (HMW) and low molecular weight (LMW) penicillin-binding proteins. HMW-PBPs are further divided into class A and class B, while LMW-PBPs are divided into four subclasses. Penicillin-binding proteins with high molecular weight are the primary target of β-lactam antibiotics. Class-A PBPs catalyze the biosynthesis of a specific linkage of monosaccharides called the O-glycosidic linkage of monosaccharides, also known as trans-glycosylation, during peptidoglycan biosynthesis [48]. PBPs contain transpeptidases or carboxypeptidases in their penicillin-binding domains, which are involved in peptidoglycan metabolism [50,51,52].

Penicillin-binding proteins play a crucial role in the biosynthesis of the bacterial cell wall’s core component, peptidoglycan. Peptidoglycan is responsible for maintaining bacterial cell strength and resisting the osmotic pressure differences between the cell wall and the external environment [53,54,55]. It forms a continuous, mesh-like structure around the bacterial cell membrane, protecting it from destruction. Peptidoglycan is a polymer composed of long glycan chains connected by a flexible peptide bridge. The glycan strands consist of repeating units of *N*-acetylglucosamine (GlcNAc) and *N*-acetylmuramic acid (MurNAc), linked by β-(1,4) glycosidic bonds. Each MurNAc residue is linked to a short amino acid chain composed of D-glutamine, L-alanine, D-alanine, and L-lysine, which are connected by pentaglycine bridges to facilitate cross-linking. Peptidoglycan biosynthesis involves the formation of stem peptides, and amino acids are cross-linked to create peptidoglycan. This transpeptidation reaction occurs in the outer cytoplasmic membrane [45,48,56,57].

### 2.1. PBP2a

Penicillin-binding proteins (PBPs), specifically PBP2 in *Staphylococcus aureus*, play a crucial role in the final step of peptidoglycan assembly (Figure 2), which is essential for the bacterial cell wall’s integrity [58,59,60]. PBP2 is a class A enzyme in *S. aureus* and is involved in the division of the cell, particularly in localizing to the septum [61].

Antibiotic use triggers a significant systemic change in the production of the mecA analog, converting it into an antibiotic resistance determinant. This determinant has also been integrated into the genetic history of *S. aureus*. Interestingly, *S. aureus* bacteria possess a mecA homolog from S. sciuri, contributing to the production of a PBP2A-like protein by *Staphylococcus sciuri* species. The mechanism by which mecA is acquired in *S. aureus* remains unknown [62].

Clinical isolates of methicillin-resistant *S. aureus* (MRSA) typically have an additional PBP2a, in addition to the four PBPs found in all staphylococcal strains. The presence of β-lactam antibiotics is essential for continuous cell wall production and bacterial proliferation [63].

The genetic architecture of the mec element downstream of the mecA gene in MRSA clinical isolates contains 13 of the most common mecA mutations, reflecting the major epidemic clones of MRSA [64].

PBP2a, the product of the mecA gene, is considered a surrogate enzyme, taking over the cell wall synthesis task from the normal complement of *S. aureus* PBPs. It exhibits transpeptidase activity, which is crucial for peptidoglycan biosynthesis [65]. PBP2a has a poor affinity for β-lactam antibiotics [66,67]. Despite its resistance to changes caused by β-lactam antibiotics, PBP2a remains a proficient catalyst for physiological processes [68]. PBP2 is the primary target for β-lactam antibiotics, and PBP2a confers resistance to all currently available β-lactam medications [69]. When interacting with β-lactam antibiotics, these enzymes undergo significant conformational shifts that may affect PBP2a’s catalytic activities in bacterial cell wall cross-linking [70].

PBP2a contributes to antibiotic resistance but relies on PBP2 for its function. The production of PBP2a is regulated by MecI and MecR1 proteins, along with the signaling proteins of the blaZ system. Three key factors required for resistance are as follows:PBP2a relies on PBP2′s transglycosylase activity.The correct length and sequence of stem peptides are necessary for PBP2a to function effectively.The presence of the pentaglycine cross-bridge is essential for PBP2a’s activity [71].

### 2.2. Peptidoglycan Biosynthesis

Peptidoglycan is a vital component that encases bacterial cells in a mesh-like structure, providing structural support and protecting them from osmotic pressure. It makes up 30–70% of the total bacterial cell wall and includes teichoic acid and lipoteichoic acid [72,73,74].

Peptidoglycan consists of alternating *N*-acetylglucosamine (GlcNAc) and *N*-acetylmuramic acid (MurNAc) units (Table 1), cross-linked by short peptides. This complex structure is crucial for maintaining bacterial cell wall stability, shape, and viability. The inhibition of PBP2a’s synthesis results in cell wall lysis and bacterial death [75].

PBPs have a catalytic site on the outside portion of the bacterial cell wall, allowing them to access their ligands without crossing the lipid bilayer. Peptidoglycan provides structural rigidity due to its highly cross-linked lattice structure. The lattice structure is formed by the alteration of two sugar units: *N*-acetylglucosamine and *N*-acetylmuramic acid, also known as pentapeptides [76].

**Table 1 molecules-28-07008-t001:** In the biosynthesis of the peptidoglycan, Mur enzymes play an essential role. Below is a list of Mur enzymes that are implicated in peptidoglycan production [77,78,79].

Sr. No.	Name of Mur Enzymes	Role of Enzyme in the Biosynthesis of Peptidoglycan
1	MurA	The enzyme MurA is responsible for catalysing the intracellular stage of peptidoglycan biosynthesis, which is the first stage in peptidoglycan biosynthesis. MurA enzyme brings such a reaction simply by transferring enol pyruvate. This enol pyruvate is produced by conversion from phosphoenolpyruvate to UDP-*N*-acetylglucosamine.
2	MurB	Reduce UDP-GlcNAc-enol pyruvate to UDP-acetylmuramic acid.
3	The ATP-dependent enzyme, Mur ligases enzymes, Enzymes from MurC to MurF	These enzymes are responsible for linking five amino acid residues. These enzymes relate to the UDP-MurNAc and thus help to form the UDP-MurNAc-pentapeptide.
4	MraY	Produce lipid I.
5	MurG	MurG adds GlcNAc moiety to lipid I resulting in the formation of lipid II.
6	MurJ	To export lipid II to the outer leaflet, MurJ switches between inward and outward conformations. Lipid II reversal is involved by MurJ [80,81].

There are three steps in the biosynthesis of the peptidoglycan, which are discussed as follows (Figure 3):

Peptidoglycan synthesis is done in a nutshell and the role of Mur enzymes is very important in peptidoglycan biosynthesis (refer ref. [82,83]. Undecaprenyl *N*-acetylmuramic acid (UDP-MurNAc) is a pentapeptide, which is then shifted to Undecaprenyl phosphate (UNDP) via the Mur Y enzyme, which works as a membrane-associated carrier. Mur G is a peripherally membrane-associated protein to which GlcNAc is coupled and is responsible for producing Lipid II [83].

**Step 1:** The nucleotide sugar-linked precursors are produced. They are, namely, UDP-*N*-Acetylmuramyl-pentapeptide and UDP-*N*-acetylglucosamine pentapeptide. The synthesis of uridyl diphosphate (UDP) acetyl muramyl pentapeptide involves acetyltransferase and uridyl transfer using acetyl Co-enzyme A (AcCoA) and UTP, respectively. From glucosamine-1-phosphate, uridyl diphosphate *N*-acetylglucosamine is formed, where Glmu catalyzes the reaction. Here, Glmu is called *N*-acetyl-1-phosphate glucosamine uridyltransferase. Peptidoglycan is a highly complex polysaccharide made up of long glycan chains that are then cross-linked by short peptide chains [84].

**Step 2:** This is also known as a lipid-linked step involving the lipid carrier undecaprenyl phosphate. Phosphor-*N*-acetylmuramyl pentapeptide and *N*-acetylglucosamine are transferred directly to the lipophilic carrier. Undecaprenyl phosphate and various disaccharides (pentapeptide)-pyrophosphate-undecaprenol are then formed. This is a reduction reaction. (Refer to Figure 3).

**Step 3:** In this step, polymerization and cross-linking of the cell wall on the cell surface occur, and the complete portion is transferred to the peptidoglycan, which is in the growing stage. Cross-bridge formation occurs simultaneously during this stage and leads to a secondary modification of the newly synthesized peptidoglycan [85].

### 2.3. Pentaglycine Bridge and Transpeptidation Reaction in Cell Wall

The composition of peptidoglycan can vary between different bacterial species. In the case of *Staphylococcus aureus*, a Gram-positive pathogen, the peptidoglycan structure is highly complex due to the presence of cross-bridges, which are synthesized by the FemXAB protein family and consist of five glycines.

The fem factor, specifically FemA and FemB, plays a crucial role in the synthesis of these pentaglycine cross-bridges in *S. aureus*. FemA contributes significantly to the second and third glycine residues, while FemB contributes to the fourth and fifth glycine residues. The pentaglycine bridge is essential for cross-linking bacterial peptidoglycans, facilitating the transpeptidation reaction that forms fibrous-like segments, which wrap around the bacterial cell wall (refer to Figure 4C).

These structural properties of peptidoglycan provide flexibility and robustness to the bacterial cell envelope, enabling it to withstand external pressure and maintain its integrity. This structural integrity is crucial for resisting forces not typically encountered by bacterial cells derived from intracellular tension and their normal state of turbidity [86].

Bacterial trans-glycosylation is a vital process responsible for polymerizing intermediates into the cell wall’s peptidoglycan. Membrane-embedded glycosyltransferase enzymes polymerize the glycan strand using disaccharides linked to lipids, including the pentapeptide precursor lipid II. The pentaglycine side chain is formed by the sequential addition of glycine residues to the membrane-bound lipid precursor (Lipid II), which is localized in the cytoplasm (see Figure 4D). FemA and FemB factors play essential roles in facilitating the generation of the pentaglycine interpeptide [87].

**Figure 4 molecules-28-07008-f004:**
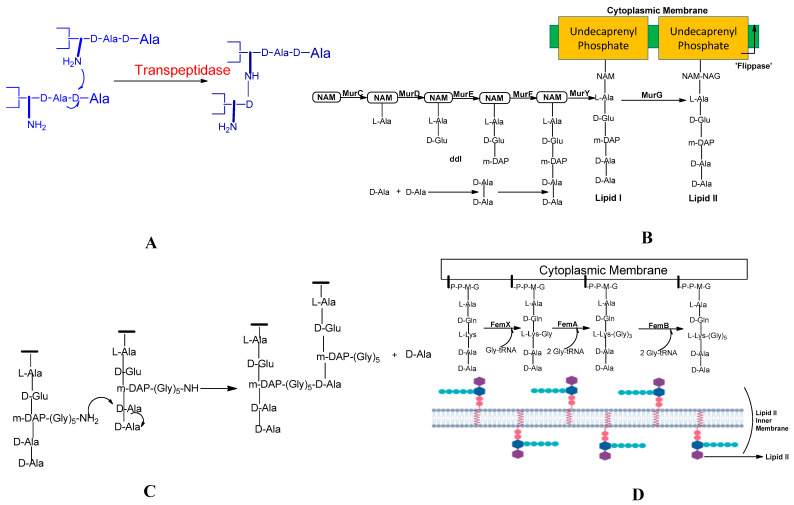
(**A**) Depiction of the transpeptidation reaction shows cross-linking within the bacterial cell wall. Ceftobiprole led to inhibition of PBP2a, and this is responsible for stopping the growth of cell wall [88]. (**B**) Illustration shows the development of the small molecule groups and disaccharide constituents of peptidoglycan in the cytoplasmic membrane. At this stage, enzymes encoded by genes successively add amino acids to the lactyl group of *N*-acetylmuramic acid (NAM), just before the complete subunit is transported across the membrane. Two amino acid residues, namely two D-alanine (D-Ala) residues, are covalently attached to produce a dipeptide and are linked as a single entity at the end of the side-chain *N*-acetylglucosamine (NAG) [89]. (**C**) Adjacent strands are cross-linked in peptidoglycan assembly by transpeptidation [90]. (**D**) Synthesis of pentaglycine interpeptide, which occurs on lipid II [91].

### 2.4. Mechanism of Methicillin Resistances in S. aureus

Epidemiological insights into Methicillin-resistant *Staphylococcus aureus* (MRSA) and *Staphylococcus aureus* infections have been derived from data collected by the Emerging Infection Program (EIP). This data has provided valuable information leading to the conclusion that MRSA is a significant infectious disease-causing bacterium.

The study involved continuous monitoring of the overall population affected by MRSA from data spanning from 2005 to 2016. Additionally, information was gathered from Premier and Cerner Electronic Health Record Databases covering the period from 2012 to 2017. These databases have been instrumental in describing global trends in both community-onset and hospital-onset MRSA and methicillin-susceptible *Staphylococcus aureus* (MSSA) infections.

According to this comprehensive study, *S. aureus* bloodstream infections have emerged as a substantial source of both mortality and morbidity in the United States. *Staphylococcus aureus* is a common pathogen responsible for a wide range of infections, affecting both individuals in the community and healthcare workers. These infections vary from superficial skin infections to more invasive and severe conditions such as sepsis, and in some cases, they can even lead to death [92].

MRSA infection can develop resistance through various mechanisms, including horizontal gene transmission of mobile genetic elements (MGEs), transposons, the presence of the staphylococcal cassette chromosome element (SCCmec) that harbors mutated mecA genes, and chromosomal mutations that affect drug-binding sites on molecular targets (refer to Figure 5) [93]. These resistance mechanisms ultimately lead to decreased drug availability for their intended targets, which is a key driver of bacterial mutation. When drug availability decreases, bacteria synthesize peptidoglycan, as the drug’s inhibition is no longer effective. Efflux proteins are also implicated in resistance by reducing drug bioavailability.

MRSA has developed resistance to all β-lactam antibiotics and other antibiotics through two main pathways:

The formation of PBPs (PBP2a): PBP2a acts as a transpeptidase and plays a crucial role in conferring resistance to β-lactam antibiotics and other antibiotics. It is involved in maintaining the cross-linking of peptidoglycan, allowing various MRSA strains to survive even in the presence of β-lactam antibiotics and other drugs.

The production of β-lactamase: β-lactamase hydrolyzes the amide bond present in the four-membered β-lactam ring, leading to the inactivation of β-lactam antibiotics, specifically penicilloic acid, before they can reach their target [94].

β-lactam antibiotics inhibit the transpeptidation reaction during cell wall biosynthesis. These antibiotics act as substrate analogs for the D-Ala-D-Ala peptidoglycan side chains, forming a long-lasting covalent acyl-enzyme complex between a serine nucleophile and the β-lactam antibiotics. The modification of penicillin-binding protein (PBP) to PBP2a is a common and critical target for cell wall inhibition. PBP2a is less susceptible to the action of β-lactam antibiotics, making it a major contributor to resistance in *Staphylococcus aureus* [95].

PBP2a catalyzes the DD-transpeptidation reaction in peptidoglycan biosynthesis (Figure 6A). Due to its reduced affinity for β-lactam antibiotics, PBP2a can effectively cooperate with trans-glycosylase to continue the biosynthesis of peptidoglycan. The expression of β-lactamase enzymes is induced when bacterial agents interact with antibiotics in the same region, and there is a metabolic cost associated with β-lactamase production [96,97].

Antibiotics, mainly β-lactam antibiotics, cause resistance simply by acquiring the mecA gene, which is mutated once, and then becomes ‘mecA encoded PBP2a (transpeptidase)’ [100]. The multi-drug efflux proteins, like QacA/B, Nora, and smr, are the proteins that are present on the cell membrane surface of the *S. aureus*. MRSA is shielded by the efflux proteins of Qac, qacA, and qacB, and ene is linked with resistance increase to non-β lactam antibiotics (refer Figure 5A). In both MSSA and MRSA, genes are resistant to various antibiotics. In penicillin, the mutated gene is blaZ. In erythromycin resistance, the responsible mutated genes are ermC and Erma. Whereas in the case of clindamycin, mainly ermC encoded inducible resistance is observed. In the case of tetracycline resistance, tetK and tetL are responsible, and in trimethoprim resistance, the mutated genes dfrA and dfrK are indicated (Figure 6B). In *Staphylococcus aureus*, the adoption of the Blaz gene generated more β-lactamase enzymes; this excess of the lactamase enzyme is responsible for the inactivation of β-lactam antibiotics, which helps them from reaching their targets [94].

### 2.5. Staphylococcus Cassette Chromosome

In the process of acquiring resistance, methicillin-susceptible *S. aureus* (MSSA) acquire mecA, a methicillin resistance gene from a mobile genetic element called staphylococcal cassette chromosome mec (*SCCmec*), leading to the formation of MRSA, which is thought to be highly infectious among staphylococcal species (*SCCmec*). On the other hand, the source of the mecA gene for methicillin-resistant *Staphylococcus aureus* is still unknown. Understanding the history of MRSA, on the other hand, necessitates locating the significance of the mecA gene. In *S. fleurettii* isan animal-related Staphylococcus species, the mecA gene has possibly originated from the *S. fleurettii* mecA gene, which was discovered on the chromosome connected to critical genes for staphylococci formation, but not to *SCCmec* [101,102]. There are some staphylococci with various peptidoglycan types. *Staphylococcus sciuri* was named after a group of staphylococci with a distinctive peptidoglycan type [103]. The staphylococcal cassette chromosome mec gene is a chromosomal genetic component, also known as a mobile genetic element, which carries the mecA gene and encodes PBP2a in methicillin-resistant staphylococci [104]. The Staphylococcus cassette chromosome mec gene (*SCCmec*) is a mutated gene and has been considered a larger heterologous mobile genetic element, which includes the central element of MRSA. SCCmec is incorporated further into the chromosome at the exact 3′ end of the orfx locus, closely located at the origin of *S. aureus* replication [105]. This plays a vital role in the immediate transcription of imported antibiotics during the initial stage, and imported antibiotics are immediately transcribed, which are then recognized as resistance genes. The ccr gene (cassette chromosome recombinase) consists of one or two site-specific recombinase genes, vital for Staphylococcal cassette chromosome mec gene (SCCmec) movement [106]. It contains three J regions. By the combination of two mutated genes, one of which is the mec gene, and the other of which is the ccr gene, a complex is produced, which is then strongly mutated; one complex is formed at the location of the insertion of the sequences, and five main gene complex types are recognized (for the types and subtypes of SCC) [107].

The mec gene complex plays a pivotal role in MRSA resistance and is encoded within the Staphylococcal cassette chromosome mec gene (SCCmec). SCCmec consists of two main components: the ccr gene complex, which encodes recombinases, and the mec gene complex, which is responsible for encoding key resistance elements [108].

The mec gene complex can be categorized into four distinct types, each associated with different resistance properties (Table 2). It is primarily responsible for the formation of PBP2a, a penicillin-binding protein with low affinity for beta-lactam antibiotics. The mec gene complex contains the mecA gene, which encodes PBP2a, and is a critical determinant of methicillin resistance [109].

The ccr gene complex is divided into three allotypes and is essential for the mobility of SCCmec. It encodes site-specific recombinases, which facilitate the movement of SCCmec elements within the bacterial genome [108,109].

SCCmec elements exhibit considerable diversity, and they are categorized into various types and subclasses. However, all SCCmec elements share two key components responsible for resistance:(a)**mecA Gene Complex:** This complex contains the mecA gene, which encodes the penicillin-binding protein PBP2 with reduced affinity for beta-lactam antibiotics.(b)**ccr Gene Complex:** This complex encodes site-specific recombinases responsible for SCCmec mobility and integration into the bacterial genome [110,111].

The nomenclature of SCCmec is established and maintained by an international working group (IWG-SCC). They have developed guidelines for SCCmec nomenclature to provide a standardized way of identifying and categorizing SCCmec elements. These guidelines help researchers and clinicians understand the structure and characteristics of SCCmec [110].

It is worth noting that identifying SCCmec elements in strains other than *S. aureus* or MRSA, especially those acquired from animals or the environment, remains a challenging task, and the best strategies for doing so are still under investigation [112].

**Table 2 molecules-28-07008-t002:** Some crucial genes are listed here, which are responsible for resistance in *S. aureus* [91,113].

Sr. No.	Name of Genes	Role in MRSA
s1	Mec (mecA, mecR1-mecIs)	The close relationship between two regulatory elements, one is *mec* and the second one is bla. BlaR2 or a close relative may be participating in mecA activation here too.
2	Chromosomal gene	“FEM” stands for factors essential for methicillin resistance. Its inactivation reduces the methicillin-resistant and *aux* (auxiliary) factors usually present in staphylococcal aureus. The majority of elements are engaged in cell wall biosynthesis and some turnover of bacteria.
3	The PBP2a operon in normal *S. aureus* contributes to resistance	The mecA is responsible for the production of PBP2a, which helps in the biosynthesis of bacteria’s cell walls. However, its role in the resistance of *S. aureus* is still unclear.
4	*Fmt*	Reduce methicillin resistance.
5	*femX*, *femAB*	When reduction of the length of the glycine side chain occurs, it is because of the inactivation of the femAB factor. The inactivation and reduced length lead to impairment in the growth of peptidoglycan and the turnover of cell-wall being reduced. All these reactions are responsible for lowering peptidoglycan cross-linking and hypersusceptibility to all β-lactam antibiotics and other antibiotics, which leads to the decrease in methicillin resistance.
6	*glnRA* (*femC*) (Mutated gene)	The precursors of muropeptide use an amino acid which is a non-amidated D-glutamate amino acid, and it takes part in the stem peptide of the transpeptidation reactions less readily, which leads to a reduction of methicillin resistance
7	*glmM* (*femD*, *femR*)	When the rate of precursor formation for peptidoglycan is reduced, it reduces methicillin resistance. It increases teicoplanin susceptibility and thus decreases methicillin resistance in MRSA.
8	Lytic enzymes (Hydrolysis Enzyme)	Murein, which is present in *Staphylococcus aureus*, causes hydrolysis. It is then needed for the growth of peptidoglycan, which is a precious cell wall content. This plays one of the leading roles in peptidoglycan growth.
9	*murE* (*femF*)	In the formation of peptidoglycan precursors, MurE necessitates the presence of this factor. This is because the abnormal precursors of peptidoglycan are present, which causes a decline in methicillin resistance; it is also possible that there may be abnormal shortening precursors involved in the peptidoglycan biosynthesis. It is still unclear.
10	*Llm* gene	The llm gene encodes lipophilic membrane proteins, affecting methicillin resistance levels and necessarily causing bacterial cell lysis rate reduction. Its functions, however, are still unclear.
11	Global regulators: *sar*, *agr*, *sigB*	The global regulators, like *sar* and *agr*, control cell density-dependent synthesis of cell wall factors that are cell wall-associated, and extracellular virulence factors. In heterogeneous MRSA, this global regulator appears to have just a slight effect on methicillin resistance in MRSA.
12	*ctaA*	Reduces the resistance to methicillin in MRSA
13	Blaz	The blaZ gene has been mutated in *Staphylococcus aureus*, and it is responsible for the breakdown of the β-lactam ring of mainly penicillin antibiotics.
14	blaI	blaI is a repressor protein that is responsible for preventing the transcription of mutated genes like blaZ or mecA.
15	blaR1	blaR1 is a transmembrane protein that signals blaZ or mecA transcription.

## 3. Structure Basis for PBP2A

The crystal structure of PBP2A, a critical factor in MRSA resistance, has been elucidated by Lim and Strynadka. This structure provides important insights into the molecular basis of its function and resistance mechanisms.

**Crystal Structure:** The crystal composition of PBP2A includes an acyl-PBP complex formed with nitrocefin, penicillin G, and methicillin. This complex allows for the comparison of the apo (unbound) and acylated (bound) forms of PBP2A. The crystal structure has a high resolution of 1.8 Å, revealing crucial details about PBP2A’s architecture [114].

**Domains and Structure:** PBP2A is a transpeptidase enzyme and is notably extensive in its structure. It comprises several key domains, including a transpeptidase loop, a non-penicillin binding region, and an elongated N-terminal domain. These domains are organized in a specific structural arrangement, contributing to the overall function of PBP2A. By deleting the transmembrane anchor region, a soluble form of the protein can be generated for characterization purposes.

**Binding Center:** The crystal structures of both apo-PBP2A and nitrocefin-acylated PBP2A revealed a unique closed conformation that cannot be approached by unmodified β-lactam antibiotics. This closed conformation suggests that an architectural shift at the binding center is required to support the antibacterial activity of PBP2A. Thus, resistance in MRSA is not solely due to the physiological reaction of PBP2a but also involves structural changes at the cell surface to counteract antibiotics.

**Active Site:** PBP2A’s active site is located in an extended groove at the N-terminus, featuring a nucleophilic amino acid, Serine 403. This site also contains a conserved oxyanion hole formed by the backbone nitrogen of Serine 403 and Threonine 600. The closed active site of PBP2A contributes to its resistance, as it hinders the binding of β-lactam antibiotics.

**Functional Role:** PBP2A can function as a single transpeptidase in cell wall synthesis, and its unique structure allows it to resist β-lactam antibiotics. It has the remarkable capability to accommodate two layers of the cell membrane, which have a substantial volume of over 1000 Å.

**Molecular Weight:** PBP2A is a high-molecular-weight class B penicillin-binding protein (PBP), specifically found in resistant bacteria like MRSA and a few other organisms, including *Staphylococcus aureus*.

The crystal structure of PBP2A (PDB ID: *1VQQ*) was resolved at 1.80 Å resolution (Figure 7). This structure, derived from a methicillin-resistant strain, provides critical structural information that aids in understanding MRSA resistance mechanisms. Researchers have identified 17 crystal structures containing ligand-bound protein structures, which contribute to a deeper understanding of the interaction between PBP2A and antibiotics [114].

This structural knowledge is invaluable for the development of new antibacterial strategies and the design of antibiotics that can effectively combat MRSA and other resistant pathogens.

### 3.1. β-Lactam as Substrate Analogues

The β-lactam antibiotics target the transpeptidase enzyme (PBP2a), which is involved in bacterial cell wall synthesis, providing mechanical strength to the bacteria. The transpeptidase enzyme catalyzes the cross-linkage between peptidoglycans and recognizes the acyl-D-Ala-D-Ala moiety as its native substrate [117]. β-lactam ring-containing antibiotics are particularly effective due to the presence of D-Ala-D-Ala groups in the cell wall. These groups serve as chain termini to which a PBP binds, forming a long-lasting suppressive covalent acyl-enzyme complex. This complex contains a nucleophilic serine group within the enzyme’s active site. The active site, which accommodates the deacylation acceptor moiety of an adjacent peptidoglycan strand or an available hydrolyzing water molecule, is typically occupied by the fused ring system of the β-lactam. Consequently, the antibiotic-bound PBP cannot undergo its usual subsequent deacylation. With the enzyme inactivated, the loss of cell wall cross-linking leads to cell lysis [118].

Ligand binding to antibiotics and peptidoglycan, or to either antibiotics or peptidoglycan, depends on substrate accessibility. Proteins adopt a structural state in the allosteric region that exposes the active site to substrate analogs. The universality of this allosteric principle has been demonstrated using multiple β-lactam antibiotics as a substrate protein of the cell membrane peptide core [119]. As we know, the β-lactam ring is the active component in β-lactam antibiotics, binding to PBPs. These antibiotics bind to the serine hydroxyl groups in the active site of bacterial enzymes, leading to irreversible inactivation by forming inactive o-acyl enzymes. This reaction demonstrates that β-lactam antibiotics containing carbonyl groups are targeted by serine residues within the active site during acylation [120].

### 3.2. About Crystal Structures for PBP2a

The crystal structure of PBP2A is documented in the Protein Data Bank (PDB) in chronological order, including the PDB ID and publication date (Table 3). Below are the PDB IDs, resolution of crystal structures, *S. aureus* species, ligands bound, and synthetic routes for a selection of ligands. Please note that only ligand-bound complexes are presented here:

PBPs, which are membrane proteins, play a pivotal role in catalyzing the transpeptidase reaction during bacterial cell wall formation. They are essential for the bacterial cell cycle and maintaining the integrity of the cell wall. Penicillin and other antibiotics disrupt bacterial cell wall synthesis by interacting with PBPs, ultimately leading to bacterial cell death, and functioning as bactericidal agents.

In the case of MRSA, in addition to the four penicillin-binding proteins (PBP 1, PBP 2, PBP 3, and PBP 4), there is a fifth PBP known as penicillin-binding protein 2a (PBP2a). PBP2a plays a crucial role in conferring resistance to beta-lactam drugs in methicillin-resistant *S. aureus* (MRSA) and is a potential target for antibacterial interventions. Notably, PBP2a and β-lactam antibiotics exhibit poor affinity for each other, leading to the ineffectiveness of many antibiotics in binding to PBP2a.

PBP2a is a unique transpeptidase enzyme responsible for conferring resistance to β-lactam antibiotics. Consequently, even if other PBPs possess transpeptidase activity, they are inhibited in the presence of these antibiotics. PBP2a can maintain peptidoglycan cross-linking despite the presence of these antibiotics [125,126].

To select compounds with antibacterial activity, the following criteria are considered:

If Pa (activity against pathogens) > Pi (activity against the indicator), the compounds are more likely to exhibit antibacterial properties.

If Pa < Pi, the compounds demonstrate reduced antibacterial activity.

## 4. Development of PBP2a Inhibitors

### 4.1. Penicillin

The challenge posed by Methicillin-resistant *S. aureus* (MRSA) extends to two fundamental classes of antibiotics: cephalosporins and penicillin. MRSA also exhibits resistance to other antibiotics, including cef-taroline, ceftobiprole, ceftazidime, cefepime, penicillin (methicillin and oxacillin), carbapenems, quinolones, and aminoglycosides, rendering it multi-drug resistant (MDR) [127]. PBP2a, a penicillin-binding protein, plays a role in MRSA’s resistance to β-lactam antibiotics, such as penicillin, cephalosporins, and carbapenems. However, MRSA remains susceptible to numerous other antibiotic agents [128].

Penicillins are chemically classified as 4-thia-1-azabicyclo-(3.2.0) heptane (Figure 8A) [129]. Tazobactam, a β-lactamase inhibitor, protects piperacillin from penicillinase hydrolysis, enabling piperacillin to bind to PBP2a and inhibit transpeptidation, thus impeding cell wall biosynthesis. Meropenem, a β-lactam antibiotic belonging to the carbapenem class, interacts with PBP1 and PBP2a at allosteric sites. Binding to PBP1 inhibits transpeptidation, while attachment to PBP2a induces active site opening. The synergistic activity of three β-lactam antibiotics has been studied in both in vivo and in vitro settings, but further research is needed to validate these findings for therapeutic use (refer Figure 8B) [130].

### 4.2. Cephalosporins

The active metabolite of ceftaroline fosamil, ceftaroline, has been approved by the US-FDA for managing acute bacterial skin and skin structure infections and community-acquired pneumococcal disease [131,132]. Ceftaroline inhibits cell wall formation by binding to penicillin-binding proteins (PBPs), with a notable affinity for PBP2a, which is associated with methicillin resistance [133,134,135]. It induces structural changes in penicillin-binding protein 2a through action on an allosteric site (PBP2a). Cephalosporins like ceftaroline and ceftobiprole operate via unique antibiotic-resistant mechanisms involving allosteric sites, leading to the inhibition of penicillin-binding protein 2a (PBP2a) [136,137]. Ceftaroline effectively targets MRSA by impacting its allosteric region. It is considered a practical and valuable therapeutic option due to its broad inhibitory activity, bactericidal effects, safety profile, and minimal adverse effects [138,139].

Reports of ceftaroline resistance in Ghana highlight the need for ongoing research and monitoring (Figure 9). The development of new anti-MRSA agents is crucial to combat infections caused by these pathogens. Cephalosporin antibiotics are characterized by a bicyclic ring structure with a four-membered lactam group, also known as cephem. Modifications to the 7-acylamino side chain have resulted in cephalosporin drugs with diverse activity patterns (ceftobiprole interaction with PBP2a–see Figure 9) [140].

CB-181963, a novel cephalosporin with an azomethine substituent at position 3 of the cephem core, demonstrates antimicrobial efficacy against MRSA. However, data regarding CB-181963’s affinity for *S. aureus* PBP2a are currently unavailable. The regulatory approval of CB-181963 will depend on comprehensive clinical effectiveness and safety profile assessments based on reported data. Further antimicrobial studies and clinical trials are warranted to establish its safety profile (Figure 8C) [141].

**Figure 8 molecules-28-07008-f008:**
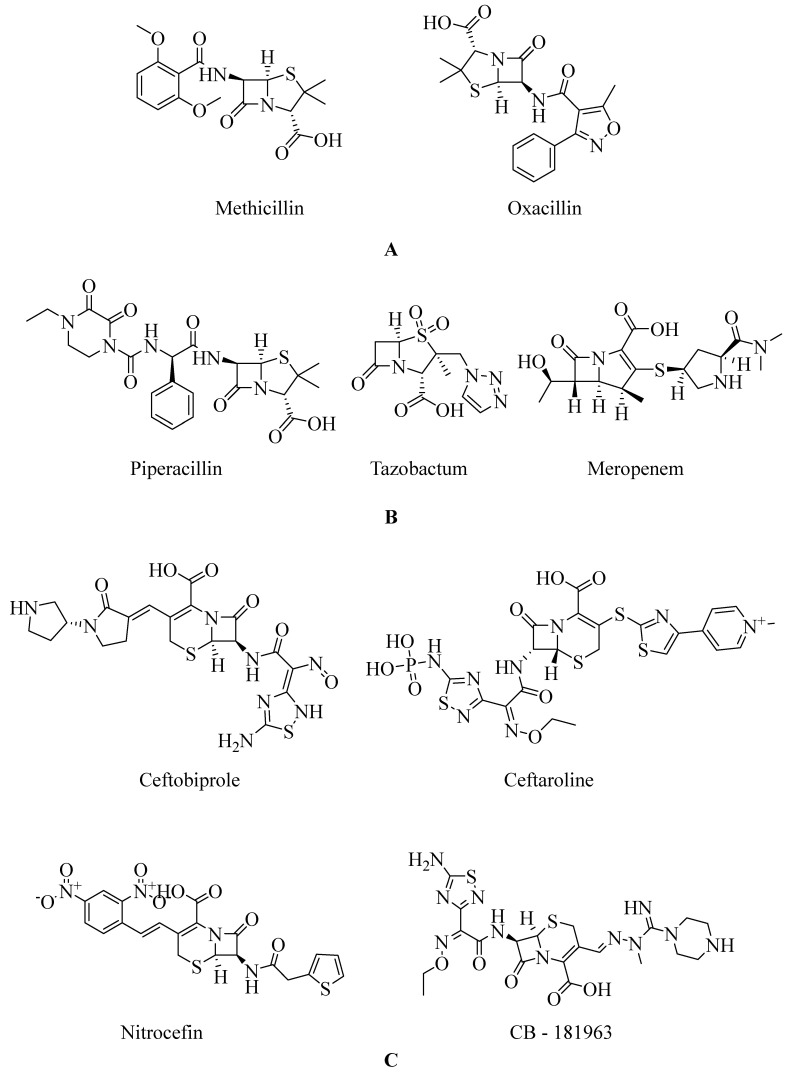
(**A**) Penicillins (methicillin and oxacillin) targeting PBPs. (**B**) β-lactam antibiotics (piperacillin, tazobactam, and meropenem) showing PBP2a inhibition when given in combination [142]. (**C**) Cephalosporins targeting PBPs (ceftobiprole, ceftaroline, nitrocefin, and CB-181963) [142,143].

**Figure 9 molecules-28-07008-f009:**
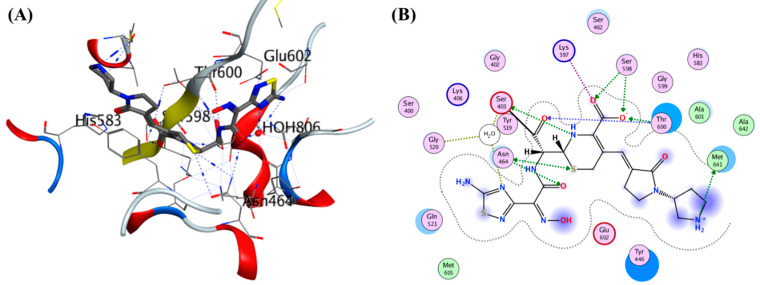
Depiction of Ceftobiprole interaction with PBP2a (PDB ID. 4DKI). (**A**) Structure shows ceftobiprole bound with PBP2a and neighboring amino acids which interact with ligand, and (**B**) showing a 2D representation of the same interaction [118].

### 4.3. Novel Pyrazole-Benzimidazole Based Derivatives

In a recent report, Yadav and Ganguly et al. [141,144] mention that a novel quinazolinone-based small molecule was found to have an inhibitory activity that was as effective as non-β-lactam antibiotics towards the MRSA pathogen by possibly targeting the PBP2a leading to the inhibition of the cell wall synthesis. The bioisosteric modifications of lead molecules helped in the retention of pharmacophoric features and led to novel quinazolinone derivative pyrazole and benzimidazole-based small molecules. On evaluation, these molecules showed moderate antibacterial activity against vancomycin and linezolid-resistant MRSA. However, these compounds provide potential leads for further hit optimization as novel antimicrobials against MRSA. These reports also mention the application of some molecular modelling studies, where pyrazole-benzimidazole was studied for its binding modes to ensure the design strategy complies with the lead compound’s reported binding mode (refer Figure 10A).

In drug discovery, benzimidazoles are heterocyclic ring systems that have attracted numerous researchers worldwide due to their therapeutic potential [144]. The benzimidazole nucleus is essential in lead discovery. Quinazolinone and benzimidazole exhibit antibacterial activity [145]. The benzimidazole ring serves as a crucial heterocyclic pharmacophore [146]. The lead compound (*E*)-3-(3-carboxyphenyl)-2-(4-cyanostyryl)quinazoline-4(3*H*), which replaces the quinazolinone core with phenyl pyrazole and benzimidazole scaffolds, is a moderately active compound, showing a significant MIC of 16 μg/mL (see Figure 10B). The synthetic scheme for structure 8 demonstrates moderate activity, marking a promising starting point for optimization to introduce a new class of PBP2a inhibitors. It may be considered a potent anti-MRSA agent [143].

### 4.4. Novel 1,2,4-Oxadiazole-Containing Derivatives

Oxadiazole was identified through in silico screening as a novel derivative of non-beta-lactam antibiotics. Lead optimization and in vivo and in vitro testing led to the discovery of antibiotics with excellent oral bioavailability and Gram-positive activity [147].

A new class of 1,2,4-oxadiazole-containing derivatives was designed by maintaining the 4-indole ring as a common feature at the C-5 position and varying the aromatic substitution at the C-3 position (see Figure 11A). Subsequently, this compound was synthesized through amidoxime and carboxylic acid heterocyclization and demonstrated synergistic activity with Oxacillin against MRSA [148,149].

### 4.5. Non-β-Lactam Allosteric Inhibitors

The acquisition of the mecA gene precedes the encoding of the PBP2a gene, which is responsible for resistance in *S. aureus* due to mutation. Preliminary modeling studies have demonstrated the inhibitory effects of reported molecules such as eMol26314565 and eMol26313223 on PBP2a, as shown in Figure 12. Subsequently, these molecules underwent in vivo and in vitro inhibition tests against antibiotic-resistant strains. ADMET investigation revealed that these compounds are non-toxic and exhibit a high rate of oral absorption. Although allosteric site inhibition against doubly mutant PBP2a was explored using quinazolinone (QNZ) pharmacophore screening, it exhibited lower binding affinity compared to the reported molecules mentioned below [148,149,150].

### 4.6. 4-(3H)-Quinazolinones

Numerous quinazolinone analogues have been proposed as potential antibacterial agents against MRSA. Molecular docking studies facilitated the development of these new quinazolinone analogues, and subsequent MM-GBSA studies were conducted. Medicinal chemistry techniques were employed to synthesize the newly designed compounds, which were characterized using IR, NMR, and HR-MS techniques. Standard drugs such as streptomycin, kanamycin, and linezolid were utilized in the biological evaluation of these compounds, which were tested against MRSA bacterial strains to investigate their resistance [152]. The MIC study of these compounds was performed using the microbroth dilution method. Compound **9**, synthesized according to the described synthetic route, exhibited significant activity with an MIC value of 31.25 μg/mL against MRSA. This MIC value was achieved by introducing a methylene linker between the amide group and the 4-methoxy phenyl substituents at the third position of the aromatic ring of quinazolinone. Additionally, the 4-chloro styryl substitution demonstrated good activity against MRSA with a MIC of ≥ 15.625 μg/mL. This research underscores how non-β-lactams can inhibit cell walls in a manner similar to β-lactams. In the future, for those interested in optimizing a lead molecule, this work will be valuable for identifying prospective lead compounds through virtual screening. In vitro assessment results indicated the effectiveness of the compounds against resistant MRSA bacterial strains, as depicted in Figure 13A, structures **1** and **2**. For the synthesis of the compound, please refer to scheme C.

Regarding the structure **3** and **4** of (*E*)-3-(3-carboxyphenyl)-2-(4-cyanostyryl)quinazoline-4(3*H*)-one antibiotics, they are 4-(3*H*)-quinazolinone antibiotics. Compound **3** exhibited an MIC value of 2 µg/mL for *S. aureus* ATCC 29213, while compound **4** had an IC50 of 63 ± 1. These antibiotics disrupt cell wall biosynthesis by binding to DD-transpeptidases involved in the cross-linking process of the cell wall, as confirmed by functional assays. For the synthesis of the compound, please refer to scheme B.

### 4.7. Pyrrolylated-Chalcones

Based on the docking results, the reported ligand (pyrrolylated-chalcone) exhibited similar bonding interactions in the PBP2a active site compared to the standard medication, chlorhexidine (CHX). Their docking scores were –7.0 kcal/mol and 8.2 kcal/mol, respectively. However, it is worth noting that no MIC study for PBP2a against MRSA was conducted.

An in vivo zebrafish model was employed to investigate various aspects, including blood vessel development, apoptosis, and normal embryonic development within a 24-h timeframe. Scanning electron microscopy analysis verifications are currently underway. The reported compound demonstrated bactericidal properties, as evidenced by the uniform shape of the *S. aureus* cells post-treatment, which transformed into irregular shapes. Moreover, there were no observed adverse effects on embryonic development.

Compound **9** (meta-hydroxy) shows promise as an MRSA inhibitor, and the zebrafish model was used to assess its in vivo toxicity. The collected data serve as a valuable starting point for hit-to-lead compound development [154], as depicted in Figure 14.

### 4.8. Bis-2-oxoazetidinyl Macrocycle (β-Lactams with 1,3-Bridges)

β-lactams with 1,3-bridges incorporated into macrocycles have been explored as potential penicillin-binding inhibitors, as illustrated in Figure 15A. During ring-closing metathesis, an unexpected dimerization resulted in the formation of bis-2-oxazetidinyl macrocycles. Some of these bis-2-oxazetidinyl macrocycles have exhibited anti-PBP2a activity.

It is worth noting that unsaturated compounds often occupy the same local minima as saturated compounds but exhibit significant alterations in their affinities. The structure of these new inhibitors is distinct from that of penicillin or cephalosporins, challenging the conventional paradigm of β-lactam antibiotics. These non-symmetrical dimers, resembling screws due to intramolecular H-bonds that impose geometrical limitations, have varying screw pitches depending on the macrocycle size, with the best results observed for 28-membered cycles. Their remarkable conformational adaptability enables them to fit into and conform to the closed conformation of the PBP2a active site.

In molecular docking, reactivity occurs through the ring opening of β-lactam, which is initiated by a nucleophilic attack at the serine active site. According to research, all of these compounds underwent testing against two high molecular weight transpeptidases, which are responsible for bacterial resistance to β-lactam antibiotics [155].

**Figure 15 molecules-28-07008-f015:**
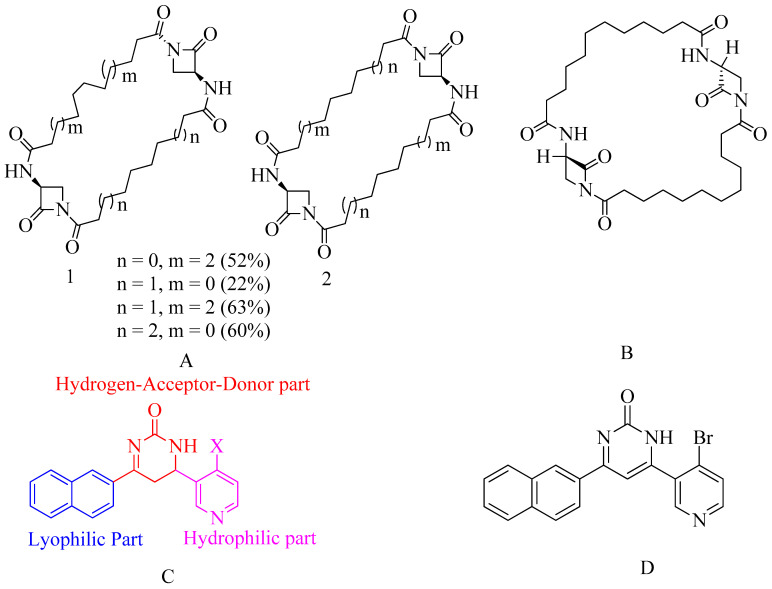
(**A**) β-lactams with 1,3-bridges inhibiting PBP2a. (**B**) Macrocycle-embedded β-lactam antibiotics. (**C**) Pharmacophore model for proposed compound. (**D**) Pyridine-coupled pyrimidinone [156].

### 4.9. Macrocycle-Embedded β-Lactams as Novel Inhibitors

Traditional antibiotics face significant challenges in forming acyl-enzyme complexes with PBP2a. However, there is a family of bicyclic β-lactam antibiotics that offers considerable flexibility in design. Among these potential inhibitors are bis-2-oxo-azetidinyl macrocycles, which consist of “head-to-head” cyclodimers of 1-(ω-alkenoyl)-3-(*S*)-(ω′-alkenoyl amino)-2-azetidinones with varying alkene chain lengths, synthesized through two metathesis reactions using the Grubbs catalyst.

In these compounds, all but one β-lactam serve as acylating inhibitors of PBP2a. The larger ring embedded within these compounds consists of 32 atoms and exhibits activity similar to that of ceftobiprole. A novel pharmacophore has been proposed based on conformational analysis, theoretical reactivity models, and docking investigations within the PBP2a cavity.

This study has identified specific compounds within this series of macrocycle-embedded β-lactam antibiotics that show the highest inhibitory activity. The compound’s calculation involves 228 atoms and 660 basis set functions in the three conserved motifs of R39. This molecular structure was generated by aligning the methanol solvent component of the SER403 model with a corresponding X-ray fragment during the docking process in PBP2a.

This research suggests a new approach to rationally designing active β-lactam antibiotics. However, further studies are necessary to assess their toxicity and ADMET (Absorption, Distribution, Metabolism, Excretion, and Toxicity) profiles [155].

### 4.10. Pyridine-Coupled Pyrimidinones/Pyrimidinthiones

In this study, in silico investigations, ADMET analysis, molecular docking, and molecular dynamics (MD) simulations were conducted on these derivatives. The objective was to validate the in vitro findings. Two potent compounds initially demonstrated promising activity, but when subjected to molecular docking experiments and MD simulations to assess both molecular stability and interactions with the target protein PBP2a, only one derivative consistently displayed superior activity.

The reported compound exhibited robust binding activity towards PBP2a with sustained stability, which was further confirmed by molecular dynamic modeling. Various advanced technologies were employed for the design, synthesis, and characterization of new pyridine-coupled pyrimidinones and pyrimidinthiones. The synthesized compounds underwent testing for their minimum inhibitory concentration (MIC), with one compound exhibiting an MIC value of 8 µg/mL. This particular compound demonstrated notable activity against MRSA (Methicillin-Resistant *Staphylococcus aureus*) infections.

The molecules are composed of three key parts:Naphthyl ring–Lyophilic unitPyridine ring–Hydrophilic unitPyrimidinone ring–Hydrogen-Donor-Acceptor (HAD) unit.

The design intentionally incorporates a lyophilic unit at one end and a hydrophilic unit at the other, with the HAD unit bridging the two ends. Among the pyrimidinone and pyrimidinthiones series, the pyrimidinone structure displayed superior anti-MRSA activity when compared to pyrimidinthiones. The choice of substituent on the pyrimidine or pyridine ring significantly influenced the antibacterial activity. Substituents at various positions were altered to investigate their impact on anti-MRSA activity. The introduction of bromo and chloro moieties, which are both halogens that create steric hindrance, resulted in enhanced anti-MRSA activity. Among these halogen compounds, bromo-containing derivatives exhibited superior activity, likely due to the stability of the bromo group.

### 4.11. Novel Naphthalimide Corbelled Aminothiazoximes

In addition to drugs like oxadiazole and flavonoids, which operate similarly to QNZ by binding to allosteric sites and inducing conformational changes that lead to active site exposure, naphthalimide, a DNA-targeting chemotherapeutic moiety, has also been investigated for its antibacterial properties. These compounds have demonstrated efficacy against MRSA.

To exploit the naphthalimide core for the development of a novel class of antibiotics affecting PBP2a, modifications were made to the fourth position of the naphthylamide core. Specifically, the aminothiazoxime group and piperazine bridges were incorporated at this position, and various substituents were introduced at the N-position. This optimization led to the creation of a compound with a remarkable MIC value of 0.5 µg/mL against MRSA (Methicillin-Resistant *Staphylococcus aureus*) (refer to Figure 16A).

The approach used in testing naphthalimide-corbeled aminothiazoximes has laid the foundation for the development of future derivatives. Compounds with a high affinity for lipase can readily attach to cell membranes. Consequently, membrane depolarization and turbulence can disrupt the cytoplasmic content, while cell membrane degradation can compromise the integrity of the cell wall. The allosteric interactions of naphthalimide-corbeled aminothiazoximes with PBP2a may expose the active site, potentially leading to the reactivation of older drugs.

A multistep synthesis process was employed to synthesize the unique naphthylamide-corbeled aminothiazoxime compound as a promising new antibacterial agent.

To synthesize the compound **4a** from the preliminary procedure for naphthalimide corbelled aminothiazoximes, the starting material used was 6-bromobenzo[*de*]isochromene-1,3-dione. To produce the naphthalimide intermediate, the coupling agents lipophilic alkylamines or acyclic amines were used. Further, reactions with piperazine and *S*-(benzothiazol-2-yl) (*Z*)-2-(2-aminothiazol-4-yl)-2-(methoxyimino) ethanethioate formed various groups of naphthalimide corbelled aminothiazoximes. In scheme B, 6-bromobenzo[*de*]isochromene-1,3-dione was used as starting material. To produce naphthalimide intermediate (See Figure 16C), the coupling agents lipophilic alkylamines or acyclic amines were used. Further, reactions with piperazine and *S*-(benzothiazole-2-yl) (*Z*)-2-(2-aminothiazol-4-yl)-2-(methoxyimino) ethanethioate formed various groups of naphthalimide corbelled aminothiazoximes. 

### 4.12. Non-Covalent Inhibitors

Anthranilic acid derivatives: screening and development of anthranilic acid derivatives show PBP2a inhibition action. Reported compounds are hypothesized (see Figure 17A) [158].

### 4.13. Investigational β-Lactam Antibiotics

A series of carbapenems were synthesized, featuring 7-acylated imidazo [5,1-b] thiazole-2-yl groups, and their bactericidal inhibitory properties were thoroughly evaluated [159]. Among these carbapenems, ME1036, originally known as CP5609, stands out. It possesses a carbapenem core with a 7-(1-carbamoyl-methyl pyridinium-3-yl) carbonyl imidazo [5,1-b] thiazole-2-yl group at the C-2 position. Clinical trials of ME1036 are currently in progress. ME1036 exhibited a strong affinity for various penicillin-binding proteins (PBPs), including PBP 2a and MF535 mec, which target PBP 1, PBP 2, PBP 3, and PBP 4. This broad-spectrum antibacterial agent displayed potent in vitro activity against methicillin-resistant staphylococci (MRSA) [159]. These findings highlight ME1036 as a promising candidate with a specific affinity for PBP2a, warranting further evaluation. Comparing two assays—a kinetic assay and an SDS/PAGE assay—it was observed that, among the reported soluble PBP2a inhibitors, L-695,256 exhibited higher potency [159]. This was determined through the filter binding assay mentioned in the literature and an electrophoretic assay employing a methicillin-resistant *S. aureus* membrane with PBP2a. The binding interactions of various carbapenems and β-lactams have been well correlated in these studies [159].

Tomopenem, another carbapenem, possesses a longer half-life compared to other carbapenems such as imipenem and meropenem. Tomopenem has demonstrated bactericidal activity against clinical isolates of methicillin-resistant *Staphylococcus aureus* (MRSA) and is currently under development in the second stage [159]. While meropenem and imipenem/cilastatin are commonly used to treat various nosocomial infections, they are ineffective against MRSA [160].

### 4.14. Carbapenem

In the case of MRSA, tomopenem exhibits a stronger binding affinity for PBP2a compared to imipenem or meropenem, both of which belong to the beta-methylcarbapenem family. This characteristic has been recognized since 2006, leading to the approval of tomopenem by the US Food and Drug Administration for the treatment of skin and skin structure infections (SSSIs) [161]. Epicatechin gallate, by delocalizing PBP2 and consequently promoting cell autolysis while restricting biofilm production, can effectively inhibit PBP2a-mediated methicillin resistance in MRSA. This mechanism makes it a promising agent in combating MRSA infections [161]. Certain alkaline boronic acids have demonstrated effectiveness against MRSA by specifically targeting PBP1 and PBP2a [161]. Corilagin, a member of the polyphenolic compound family, has shown potent antibiotic activity against MRSA. Its mode of action involves interfering with the formation of bacterial cell walls, with a primary impact on PBP2a activity [162]. 

**Figure 17 molecules-28-07008-f017:**
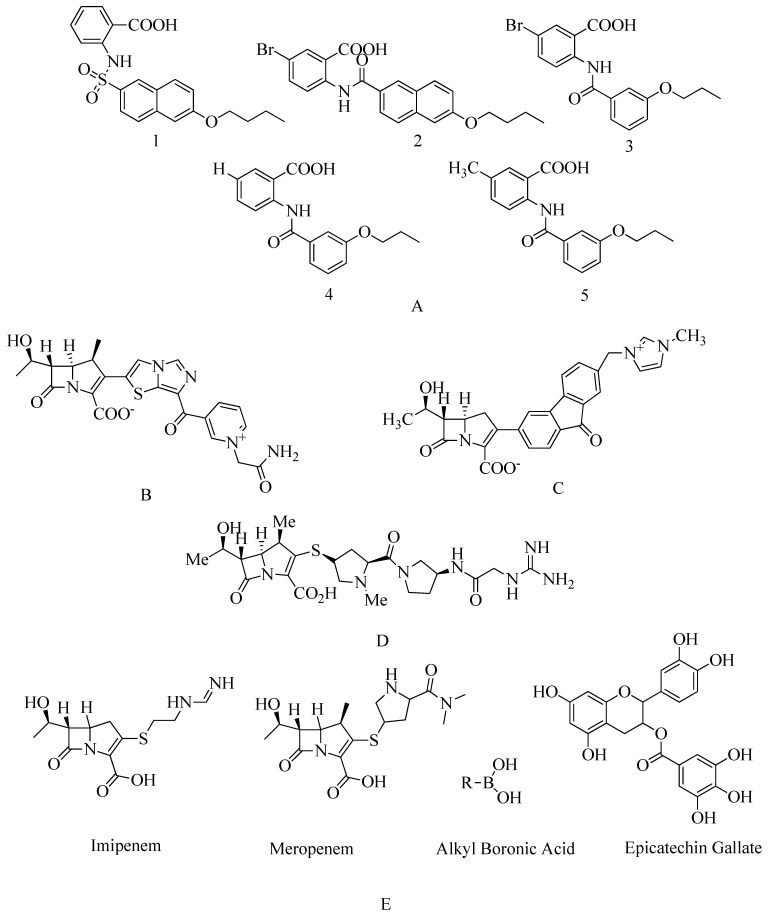
(**A**) Non-covalent inhibitors. (**B**) Novel carbapenem ME1036 [163]. (**C**) Novel carbapenem L-695,256. (**D**) Tomopenem. (**E**) Structure of carbapenem, flavan-3-ol, and alkyl boronic acid shows inhibitory action against PBP2a.

### 4.15. Novel Benzoxazole Derivatives

Adenine and guanine are elemental structural isotopes and essential heterocyclic compounds found in nucleic acids. These compounds form benzoxazole rings, and it is believed that derivatives containing this ring structure may possess the ability to inhibit microbial growth by interfering with nucleic acid synthesis.

One such derivative, studied for its antibiotic properties, is 2-(p-tert-butylphenyl)-5-(3-substituted propanamido) benzoxazole (see Figure 18A). This substance has been thoroughly evaluated and has demonstrated greater effectiveness compared to reference tablets, with an MIC value of 8 µg/mL against MRSA and VREF. The interaction between this compound and the allosteric site of PBP2a, as well as the active site of PBP4, has been explored in 2D and 3D connections, highlighting its potential antibacterial activity.

### 4.16. Pyrazolylpyridine Analogues

The effectiveness of the compound was confirmed by assessing its impact on MRSA cellular membranes. Specifically, the damage to the MRSA cell membranes was monitored by observing potassium flow and the release of cellular content, including DNA. The data revealed that as the drug dosage increased, the rate of DNA leakage also increased, indicating damage to the MRSA cell membrane.

Furthermore, research indicated that compounds **1** and **2** interact with amino acid residues such as HIS293, VAL277, TYR446, HIS583, and THR582 at the active and allosteric sites of PBP2a. Spectroscopic techniques, including FTIR, NMR, mass spectrometry, and molecular docking, provided evidence that compound **2** effectively targets PBP2a. Moreover, compound **2** exhibited the ability to inhibit biofilm formation, which is a protective mechanism employed by MRSA. It achieves this by acting on both the active and allosteric sites of PBP2a, contributing to its effectiveness against MRSA. These antibacterial findings were further validated through in silico studies.

Combination therapy involving compound **2** has shown promising results and has the potential to prevent MRSA bacteria from developing resistance. The MIC value of compound **2** was determined to be 30 ± 0.4 μg/mL (see Figure 19A).

### 4.17. Miscellaneous PBP2a Inhibitors

#### 4.17.1. Flavonoids

Flavonoids, which belong to the plant kingdom and are considered secondary metabolites, are a class of chemicals found in plants. They are particularly promising as potential agents against MRSA, especially non-glycosylated types of flavonoids. Interestingly, a combination of glycosylated derivatives, such as flavonolignan silibinin A, has been found to enhance the inhibitory action of ampicillin against MRSA. To understand the mechanism of action of flavonoids in combating MRSA, an extensive inverse virtual screening (VS) approach was employed. In this approach, penicillin-binding protein 2a (PBP2a) played a crucial role as a mediator in the synergistic antibacterial activity of flavonoids. Many of these flavonoids are phenolic compounds, including hesperidin and its aglycon hesperetin, Rutin, and aglycon quercetin. Among the various flavonoids subjected to virtual screening, polyhydroxylated flavonoids have shown promising synergistic activity against PBP2a (see Figure 20A). Hesperetin, identified as (S)-5,7-dihydroxy-2-(3-hydroxy-4-methoxyphenyl) chroman-4-one, stands out as a flavonoid with potential for further in vitro and in vivo evaluation. Methicillin is commonly used for therapeutic testing against MRSA (refer Figure 20A) [166]. 

#### 4.17.2. Demethoxycurcumin

The goal of this trial was to see if DMC could prevent MRSA from causing pathogenicity. A set of drug susceptibility assays were conducted to determine DMC’s inhibitory effect on various MRSA strains [167]. Synergistic interactions with β-lactam antibiotics were discovered, as were several essential resistance genes. Demethoxycurcumin (DMC [167], see Figure 20B) shows a significant effect in treating MRSA in holistic treatment-resistant bacteremia. Research on dimethoxycurcumin is expected to continue in the future. DMC will likely play a role in combating MRSA. The checkerboard method and time-kill experiments have been used to detect the action of antibiotics and the synergistic action of dimethoxycurcumin. The reported data show that the MIC value of dimethoxycurcumin was 62.5 µg/mL [167]. At a 250 µg/mL dose, curcumin was found to reduce PBP2a levels in MRSA. When combined with oxacillin, such protein deficiency was confirmed This decrease in PBP2a suggests that curcumin may act by interfering with translation or transcription activities. Its synergistic effect with oxacillin completely reduced PBP2a levels compared to the synergistic effect of curcumin [168].

#### 4.17.3. Quercetin 3-o-Rutinoside

The allosteric and active sites of PBP2a were found to have a better binding affinity for Quercetin 3-o-Rutinoside and Ceftaroline, which could be beneficial for combination therapy. When cephalosporins such as cefixime and ceftriaxone are combined with quercetin 3-o-rutinoside, they show a reduction in MRSA activity [169]. When used alone, they are less effective than when used in combination. This combination induces morphological changes on the surface of the bacteria. The effectiveness of the combination of rutinoside and cefixime is comparable to that of cefixime and rutinoside on a single bacterial surface and cell wall [169,170]. Overall, these findings suggest that rutinoside has promising potential for MRSA combination therapy. However, further research is needed to confirm the effectiveness of rutinoside in synergistic strategies and its role as a PBP2a modulator against MRSA infection [169,170]. Additional studies are required to establish quercetin 3-o-rutinoside as a PBP2a inhibitor [169,170].

#### 4.17.4. Ursolic Acid 3-*O*-α-l-arabinopyranoside with Oxacillin

Oxacillin is a β-lactam antibiotic that functions by inhibiting the biosynthesis of cell wall peptidoglycans. It does so by competitively binding to and inhibiting penicillin-binding proteins (PBPs). The synthesis of PBP2a, which is conferred by the mecA gene, leads to resistance in *S. aureus*. PBP2a is a protein with a low affinity for β-lactam antibiotics. PBP2a disrupts the normal functioning of PBPs by competitively blocking their activity. Therefore, one effective approach to restoring antibiotic sensitivity in MRSA is to inhibit or reduce the expression of PBP2a [171] (see Figure 20C).

#### 4.17.5. Thioridazine

Thioridazine is utilized for treating various diseases caused primarily by antibiotic-resistant but gram-positive microorganisms like MRSA. Thioridazine, when exposed to laser radiation, reproduces itself and produces photoproducts, offering a rapid and cost-effective development method compared to traditional synthesis methods. When MRSA membrane proteins are exposed to radiation in a thioridazine solution, it can have a potentially adverse effect, leading to the inhibition of PBP2a expression by sulforidazine and mesoridazine [172] (see Figure 20D).

#### 4.17.6. Metronidazole-Triazole Hybrids

MRSA resistance was observed in all the drugs that were developed. Oxacillin, the largest protective molecule, exhibited good coordination with the reference drug. These compounds were found to be non-toxic to the THP-1 cell line at concentrations of up to 50 M. Additionally, these molecules demonstrated suitable pharmacokinetic parameters and adhered to Lipinski’s rule of five. Consequently, researchers have suggested that these compounds could serve as a starting point for the development of new anti-MRSA drugs. The synthesis of 1,4-disubstituted 1,2,3-triazole compounds, which are bioisosteres of amide bonds, was achieved through copper-catalyzed azide-alkyne cycloaddition. One of these compounds exhibited effectiveness with an MIC value of 4 µg/mL against nine MRSA strains [173] (refer Figure 20E–G).

#### 4.17.7. Aspermerodione

Small agents that attach to the allosteric protein surface of PBP2a may potentially increase the susceptibility of MRSA to conventional β-lactam antibiotics. Aspermerodione, a novel complex terpene-polyketide hybrid with a unique carbocyclic core, is one such agent. It damages the bacterial cell wall or cell membrane and may act as a PBP2a inhibitor, working in conjunction with oxacillin and piperacillin to provide antibacterial activity against MRSA when used with β-lactam drugs. Aspermerodione features a spiro [bicyclo [3.2.2]nonane-2,1′-cyclohexane] ring system and a 2,6-dioxabicyclo [2.2.1] heptane group [174] (see Figure 20H).

#### 4.17.8. Chitosan-Ferulic Acid Conjugates

Chitosan is a naturally occurring mucopolysaccharide known for its biodegradability and biocompatibility, with low toxicity. Research has explored the in vitro bactericidal effectiveness of chitosan-ferulic acid compounds in combination with other antibiotics against MRSA. However, further studies are needed to gather additional data on their safety profiles and their potential to inhibit the mecA gene and PBP2A synthesis. These compounds have demonstrated the ability to reduce the expression of the mecA gene in a dose-dependent manner [128] (See Figure 20I).

#### 4.17.9. Indole-Nitroimidazole Conjugates

Penicillin-binding protein 2a (PBP2a) is a protein found in MRSA that plays a crucial role in completing cell structure cross-linking and reducing susceptibility to β-lactam antibiotics. Bioassays have indicated that this compound exhibits a significant inhibition of MRSA, with an MIC value of 0.1 µg/mL [20] (see Figure 20J–L). It binds to PBP2a and reduces the expression of resistance-related genes in MRSA.

#### 4.17.10. Polyphenol

Tellimagrandin II is a polyphenol extracted from the Trapa bispinosa plant, specifically from the shell of this plant. It has been found to inhibit the regulation of de novo mutations in MRSA, reduce mecA transcription, and inhibit the activity of PBP2a. Tellimagrandin II works by reshaping bacteria and inhibiting PBP2a-mediated resistance, effectively preventing the development of MRSA and MSSA strains. It has a minimum inhibitory concentration (MIC) of 128 g/mL [20] (refer Figure 20M), making it a promising new anti-Staphylococcus agent. The antibacterial effect of tellimagrandin II against MRSA not only reduces the required dose of current antibiotics but also helps overcome drug resistance in *S. aureus*.

**Figure 20 molecules-28-07008-f020:**
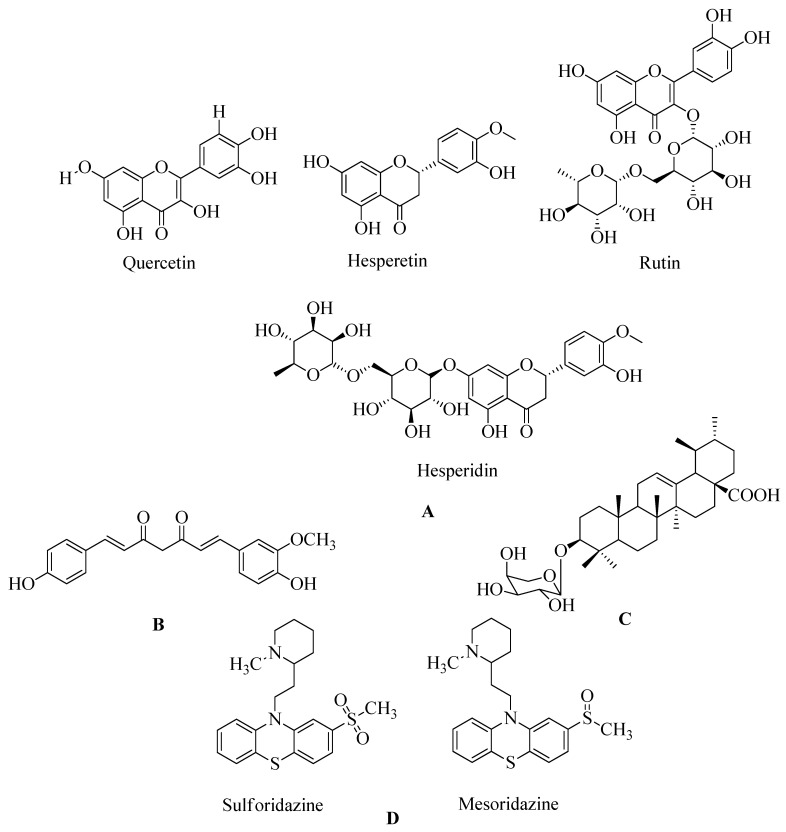
(**A**) Structure of polyhydroxylated flavonoids showing promising synergistic activity against PBP2a [175]. (**B**) Structure of demethoxycurcumin [167]. (**C**) Ursolic acid 3-*O*-α-l-arabinopyranoside structure. (**D**) Structure of Thioridazine. (**E**) Formation of hybrid. (**F**) Structure of Metronidazole-Triazole Hybrids. (**G**) Synthetic route for Metronidazole-Triazole Hybrids. Reagents used in synthesis and procedure: reagent (a) p-TsCl and Pyridine, (b) NaN3 and DMF (c) t-BuOH: H_2_O (1:1), Sodium ascorbate, CuSO4.5H2On used for the synthesis of metronidazole-triazole hybrids [173]. (**H**) Structure of Aspermerodione. (**I**) Synthesis route of chitosan-ferulic acid conjugate. (**J**) Indole-Nitroimidazole conjugates [176]. (**K**) Synthesis route of Indole-Nitroimidazole conjugates; (a) potassium carbonate, (b) dimethylformamide, and a phosphorus oxychloride catalyst, promoters, piperidine, and glacial acetic acid [176]. (**L**) Indole-Nitroimidazole conjugates intermediates. (**M**) Structure of Tellimagrandin II [177].

## 5. Screening Technologies

There are several screening methods available, such as computer-aided screening, High Throughput Screening (HTS), pharmacophore-based screening, inverse virtual screening (VS), reverse virtual screening (RVS), and Surface Plasmon Resonance (SPR) assay, for the discovery of novel PBP2a inhibitors [174]. The following methods and in silico techniques have already been used to identify various types of PBP2a inhibitors.

### 5.1. High throughput Screening

High throughput screening involves the exposure of a large number of compounds to identify potential hit compounds, wherein a vast array of chemical compositions is compared against a defined target. The advantages of HTS over other screening techniques include its efficiency within a short timeframe. This method yields favorable results, employs simple techniques, is cost-effective, demonstrates excellent effectiveness, and has the capacity to generate high-quality data on ligand-target interactions. Virtual high throughput screening (VHTS) was conducted for carbapenem derivatives. In the past century, carbapenems and penicillin-binding proteins were employed as blockers for PBP2a. Thienamycin stands as the first natural carbapenem derivative. Thousands of thienamycin derivatives have been formulated, with 273 non-toxic chemicals earmarked for future research. Molecular dynamic simulations were performed following docking to ensure the stability of the pharmacophores. Only a select few exhibited penicillin-binding protein 2a inhibition activity. Based on docking analysis, five compounds—imipenem, meropenem, doripenem, ertapenem, and thienamycin—were chosen as leads. The results indicated that nearly all molecules interacted with the reference ligands, unveiling the active site of PBP2a through proteomics research. In comparison to known inhibitors, imipenem and meropenem exhibited similar actions against PBP2a (see Figure 21).

### 5.2. Computer-Aided Screening

Virtual screening is widely employed for identifying new inhibitors for specific targets [179]. Computer-assisted drug design serves as a starting point for discovering novel compounds, utilizing data crucial in the field of drug discovery. ADMET (Absorption, Distribution, Metabolism, Excretion, and Toxicity) and docking experiments are conducted to assess drug-similarity and evaluate the effectiveness of various herbal substances. Computer-aided drug design techniques play a pivotal role in exploring potential therapeutic agents, streamlining drug development processes to overcome numerous challenges. The process of identifying lead molecules with robust pharmacological properties and drug-like characteristics is typically time-consuming. Computer-aided technology simplifies the identification of chemically active molecules with favorable ADMET profiles. To optimize results, data from in silico research are subsequently utilized in in vitro tests, selecting chemical compounds with potent pharmacological and druggable properties. The computer-aided approach offers a rapid and efficient screening process. Out of the 74 compounds derived from medicinal plants, only a handful were identified as superior candidates against MRSA. Lead molecules, such as meliantriol and related compounds, exhibit potential effectiveness surpassing that of vancomycin. As per computer-aided research, plant-derived inhibitors hold considerable therapeutic value and warrant in vitro testing. Computer-assisted ADME testing and toxicity estimation models were the preferred methods for screening the best ligands with robust pharmacological activity. Herbal substances containing meliantriol, β-Sitosterol, and related compounds have demonstrated potent inhibitory effects against MRSA PBP2a [180].

### 5.3. In Silico Docking Screening

In silico screening was conducted to identify new compounds that target specific PBP2a allosteric sites under acidic conditions. Various chemical databases, including eMolecules, ChEMBL, and ChEBI, were examined for potential inhibitors resembling quinazolinone. The PBP2a binding mode was calculated using molecular docking, utilizing a crystallized ligand with quinazolinone as a reference. A total of 35 compounds underwent virtual screening, with only two compounds showing promising inhibitory activity based on their binding affinity. These two compounds, eMol26313223 and eMol26314565, exhibited potential for the in vitro and in vivo suppression of the mutant PBP2a. Additionally, their efficacy was confirmed through tests involving antibiotic-resistant bacterial strains. Post-dynamic assessments indicated that both blockers of the mutant PBP2a displayed high stability and ligand binding [151].

The representation of PBP2a and its ligands, as per the data in Figure 22A, revealed the X-ray composition of *N*-acetylmuramic acid (NAM)-pentapeptide (medium, colored according to molecular type; yellow carbon) at one end of the allosteric site cavity. The ligand’s NAG moiety was also absent in the electron density due to its dynamics. When peptidoglycan attaches to the allosteric site, the active site opens, accommodating the two peptidoglycan strands for the cross-linking process. The resulting cross-linked peptidoglycan (colored by atom types; carbon in dark grey) is displayed at 1 o’clock, linked to the active site as the outcome of the reaction, as reported by the researcher. After the enzyme undergoes a 90-degree rotation on the y-axis in the viewer’s direction, the extension offers a stereo image of the cross-linked peptidoglycan attached to the underlying chemical structure. Quantum mechanical/molecular mechanical (QM/MM) calculations were employed to determine the structure of the bound cross-linked peptidoglycan.

Following in silico docking results, the discovery of (*E*)-3-(3-carboxyphenyl)-2-(4-cyanostyryl)quinazoline-4(3*H*)-one as an effective antibacterial agent against methicillin-resistant *S. aureus* (MRSA) was achieved through the screening and chemical synthesis of a collection of quinazolinones. This family of antibiotics has demonstrated effectiveness in treating MRSA infections. The compound exhibited an IC50 of 63 ± 1 μg/mL [153]. Antibiotics have shown promising effects in combating MRSA infections. In particular, three compounds—berbamine, hypericin, and galangin—displayed the highest binding affinity (kCal/mol) of −11.5, −8.3, and −8.3, respectively, against the essential MRSA protein known as PBP2a (transpeptidase) [176].

### 5.4. Multiple Virtual Screening Techniques

Inhibitors targeting the penicillin-binding protein 2a (PBP2a) of *S. aureus* have been identified as a significant breakthrough. PBP2a plays a crucial role in conferring methicillin-resistant properties to *S. aureus*, making it a potential target for numerous antibiotics. This study employed a hybrid virtual screening approach, combining drug-likeness criteria with biological assessments, to uncover novel medications. PBP2a inhibitors represent a relatively recent class of drugs. The docking techniques employed include LibDock or CDOCKER docking, along with various docking-based assays. All candidates were rigorously evaluated for compliance with Lipinski’s rule of five and ADMET criteria. Through the virtual screening of commercially available chemical compounds, new PBP2a inhibitors were discovered. Eleven promising compounds were selected from the final hits and are currently under laboratory investigation. Notably, Hit 1, Hit 2, and Hit 3 exhibited excellent anti-MRSA activity against ATCC 33591 with low toxicity (refer to Figure 23).

The affinity of the three drugs for PBP2a was determined using Surface Plasmon Resonance (SPR) measurements in conjunction with molecular dynamics (MD) simulations. Through the study of inter-complex interactions, it was observed that all of the hit compounds effectively optimized the allosteric sites of the PBP2a protein. Notably, Hit 2 exhibited the best binding affinity (KD = 1.29 × 107 M) and substantially inhibited MRSA clinical isolates. These findings indicate that these compounds, especially Hit 2, have the potential to serve as effective non-beta-lactam antibiotics against MRSA. Further research on these hit compounds may pave the way for the development of specific drugs that target the interaction with PBP2a, offering new avenues for MRSA treatment in the future [125]. 

### 5.5. X-ray Crystallography

Structure-based drug discovery, including fragment-based drug discovery, benefits from techniques such as X-ray crystallography. X-ray crystallography is a powerful method that provides comprehensive insights into the three-dimensional structure and interactions between small molecules and their target macromolecules. It is a valuable tool for screening fragment libraries and confirming potential “hit” compounds identified through various screening techniques against target macromolecules. The success of X-ray crystallography-based ligand screening depends on the quality and nature of the crystal structures used in the study [178].

Ceftobiprole is a noteworthy example of a β-lactam antibiotic with anti-MRSA activity. It targets PBP2a, a key factor in β-lactam resistance in MRSA. Ceftobiprole is effective against MRSA, Enterococcus faecalis, and Streptococcus pneumonia. Its unique structure, featuring the oxyimino aminothiadiazolyl group (R1 group) attached to the 7-amino group of the cephalosporin nucleus and the vinyl-pyrrolidinone moiety (R2 group) at position 3, enhances its interaction with the PBP2a active site. This interaction disrupts PBP2a’s function and combats MRSA [118].

Ceftaroline is another recent β-lactam antibiotic that inhibits PBP2a by inducing an allosteric conformational change that exposes the active site [179]. Typically, β-lactam antibiotics irreversibly acylate the active site serine in susceptible bacteria’s PBPs, resulting in the loss of transpeptidase activity. However, PBP2a, with its low binding affinity to β-lactam antibiotics, allows MRSA to continue cell wall production even in their presence. Understanding the structural basis of PBP2a resistance and the features of newly discovered lactams that are critical for high binding is crucial for developing effective MRSA treatments. A comparative analysis of PBP2a’s active site in different forms, such as apo and acetylated complexes with nitrocefin, penicillin G, and methicillin (PDB ID-1VQQ, 1MWS, 1MWT, 1MWR, and 1MWU), provides valuable insights into this aspect [115].

### 5.6. Miscellaneous Method of Screening

#### 5.6.1. Inverse Virtual Screening

Flavonoids, which are secondary metabolites derived from plants, represent a promising class of compounds for combating MRSA infections. This class of compounds includes non-glycosylated flavonoids. In combination with flavonolignan silibinin A, glycosylated derivatives have been shown to restore the inhibitory effect of ampicillin against MRSA. To understand the mechanism of action of flavonoids against MRSA, researchers conducted comprehensive inverse virtual screening (IVS). This screening identified penicillin-binding protein 2a (PBP2a) as a potential target responsible for mediating the antibiotic-synergistic activities of flavonoids in this class. Subsequent to the IVS, further investigations involved in vitro studies, molecular docking, molecular dynamics simulations, and the analysis of PBP2a binding modes [175].

#### 5.6.2. Reverse Virtual Screening

In order to assess the antibacterial mechanism of the icariin derivative shown in Figure 24A,B, reverse virtual screening was conducted. This screening process aimed to identify the compound’s potential interactions with specific biological targets. Surface plasmon resonance (SPR) assays were used to confirm the binding affinity of the icariin derivative with PBP2a, a key protein involved in cell wall synthesis. The study revealed that this compound exhibits a high binding affinity for PBP2a, suggesting its ability to inhibit cell wall synthesis through interactions with this protein [180].

#### 5.6.3. Microtiter Plate-Based Assay

High levels of β-lactam resistance in methicillin-resistant *S. aureus* (MRSA) are often attributed to mutations in PBP2a (penicillin-binding protein 2a). To identify potential PBP2a inhibitors, several screening methods are employed. One such method is the microtiter plate-based assay, which proves to be an effective and convenient approach for the development and characterization of new inhibitors specifically designed to target PBP2a.

This assay involves the description of PBP2a inhibitor binding, primarily based on the binding of radio-labeled β-lactams. Various penicillins, cephalosporins, carbapenems, and their derivatives are utilized in this study to assess the binding of these molecules to the PBP2a target [181].

#### 5.6.4. Ab Initio Methods of Screening

The development of large ring 1,3-bridged 2-azetidinones, which are the parent structures of carbapenems (with 2-hydroxyethyl side chains on C3 and thienamycin stereochemistry), as well as the cephalosporin family (with 2-hydroxyethyl side chains on C3 and thienamycin stereochemistry), and the cephalosporin family (with 2-hydroxyethyl side chains on C3 and thiamine stereochemistry) is carried out using ab initio screening techniques. This research aims to determine the binding affinity of PBP2a inhibitors and identify new hits for MRSA infection. Out of the twelve compounds studied, only one showed good to excellent inhibition activity, while two exhibited modest activity against PBP2a [182].

## 6. Conclusions

MRSA, as a multi-resistant and pandemic pathogen, poses a significant threat to public health, and there is a pressing need for new anti-MRSA compounds to combat drug resistance. PBP2a, a key enzyme involved in bacterial cell wall formation, is an attractive target for preventing cell wall production and has shown promise in anti-MRSA drug development.

A variety of screening techniques, including high throughput screening, virtual screening, computer-aided design, molecular docking screening, X-ray diffraction screening, and other methods such as ab initio screening, reverse-virtual screening, inverse-virtual screening, and microtiter-plate based assays, have been employed to identify PBP2a inhibitors. These efforts have led to the discovery of several classes of PBP2a inhibitors, each with its unique scaffold and mechanism of action.

The enzymes involved in *S. aureus* cell wall biosynthesis are also promising targets for MRSA drug discovery, particularly those that inhibit PBP2a. Focusing on both pharmacokinetics and pharmacodynamics can help improve the in vivo properties of these drugs. The development of PBP2a inhibitors holds promise for combating antibiotic resistance in MRSA. Combining novel agents in future treatments may yield effective results.

In summary, addressing the urgent need for effective anti-MRSA compounds is crucial, and ongoing research efforts are making significant strides in this direction by targeting PBP2a and other enzymes involved in cell wall biosynthesis.

## 7. The Outlook

The generation of peptidoglycan precursors is a critical process in methicillin-resistant *Staphylococcus aureus* (MRSA).Discovering novel medicines against MRSA is often better and more convenient using a ligand-based method.The main challenges in creating novel anti-MRSA agents include: (a) How to leverage existing knowledge to develop anti-virulence medications. (b) How to integrate these findings into the evolving field of bacterial interactions within the human microbiome.Utilizing knowledge from various articles is crucial for discovering new antibiotics to combat MRSA infections. As antibiotic resistance poses a significant threat to public health, it is essential to improve existing antibiotic classes or explore entirely new classes of antibiotics (Figure 25) [183].

## Figures and Tables

**Figure 1 molecules-28-07008-f001:**
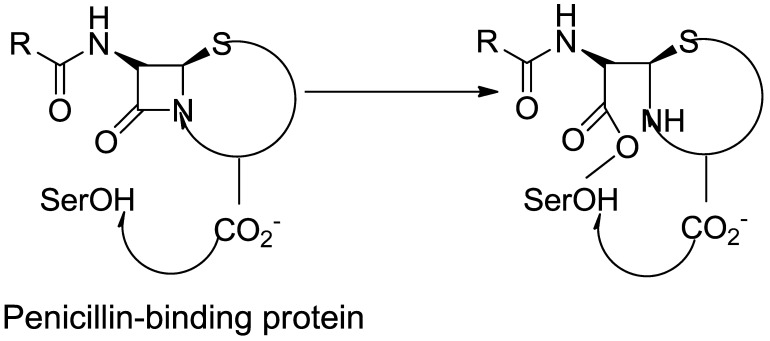
Penicillin-binding proteins are susceptible to β-lactams [45].

**Figure 2 molecules-28-07008-f002:**
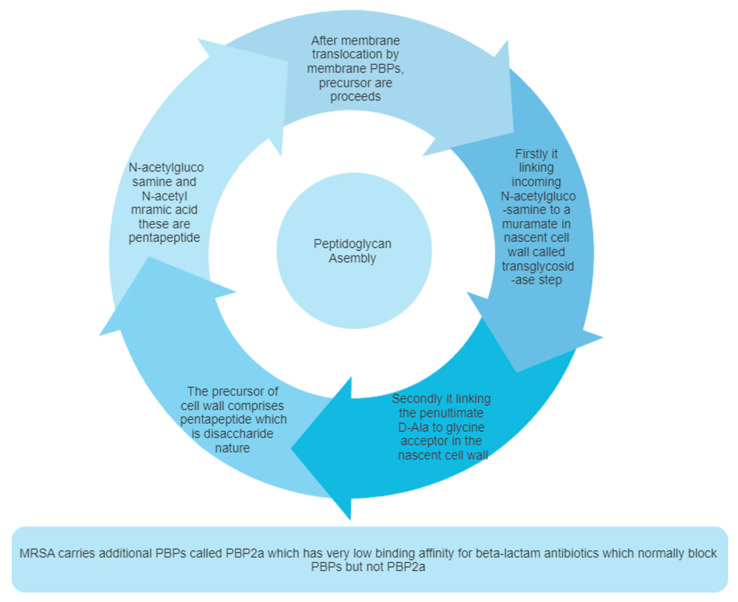
Cell wall assembly [58,59,60].

**Figure 3 molecules-28-07008-f003:**
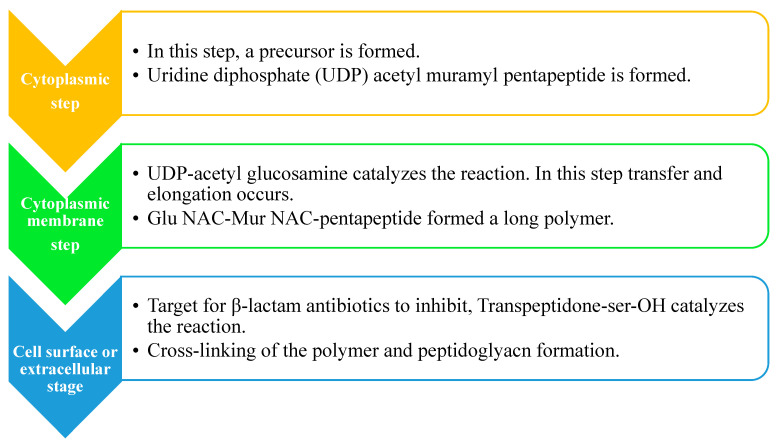
Steps in the biosynthesis of peptidoglycan.

**Figure 5 molecules-28-07008-f005:**
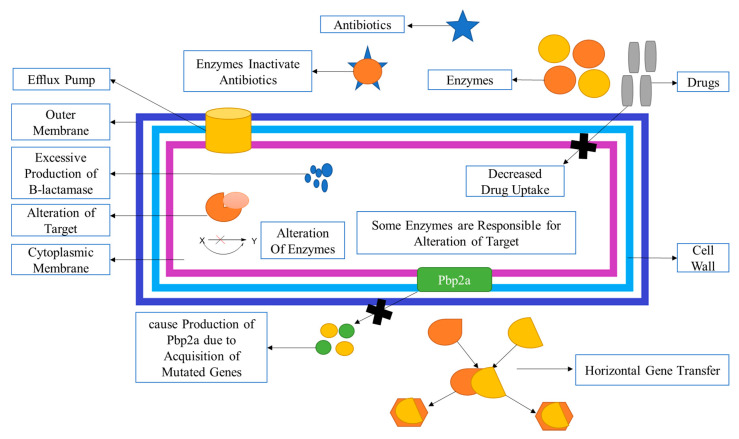
A Detailed illustration of MRSA resistance. The large rectangular box with three layers refers to the cell wall, components on the three-layered box represents the component belonging to the cell wall. Star shaped structures are the antibiotics and the enzymes are presented as round structures.

**Figure 6 molecules-28-07008-f006:**
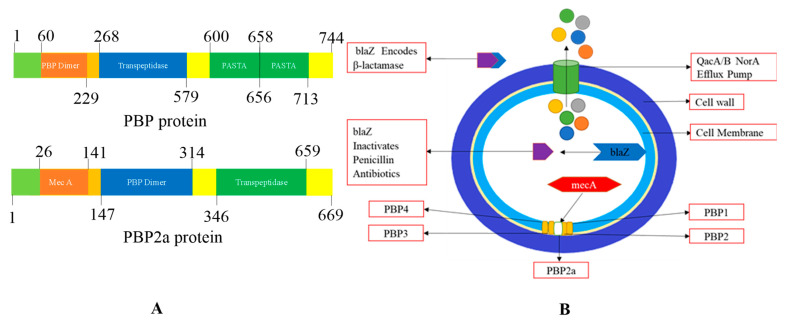
(**A**) The domains of penicillin-binding protein (PBP) and penicillin-binding protein 2a (PBP2a) of the target strain are depicted in the diagram [98]. (**B**) Schematic representation of resistance in *Staphylococcus aureus*. β-lactamase is a bacterial enzyme. This enzyme breaks down the β-lactam ring, which makes antibiotics useless before they contact the PBP target [99].

**Figure 7 molecules-28-07008-f007:**
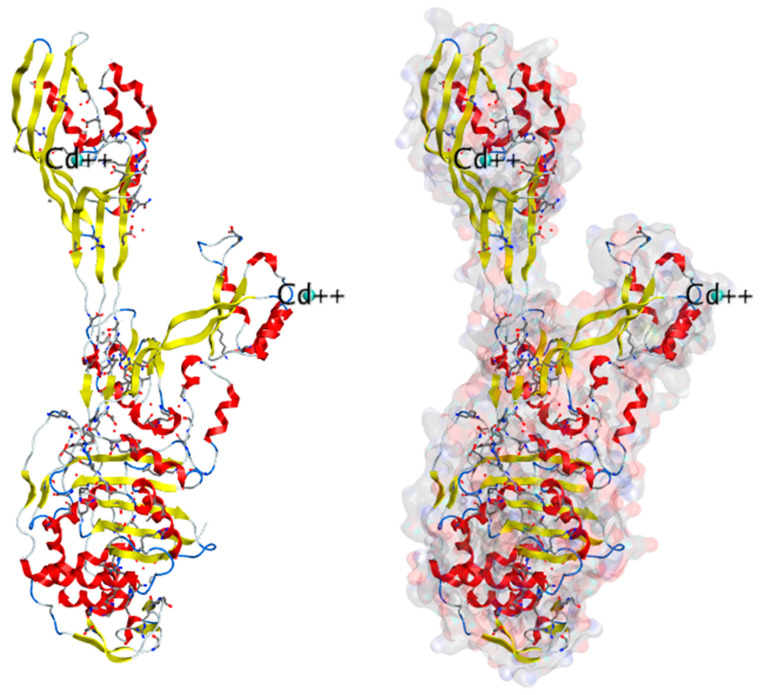
The surface interface of PBP2a (PDB ID. 1VQQ). It is a soluble structure of MRSA strain 27r, which is determined at 1.80 Å resolution. This provides the highest resolution for high molecular mass PBP [115]. The active site residues of penicillin-binding protein 2a (PBP2a) are Ser403, Lys406, Arg445, Tyr446, Glu447, Ile459, Glu460, Ser403, Ser462, Asp463, and Asn464 [116].

**Figure 10 molecules-28-07008-f010:**
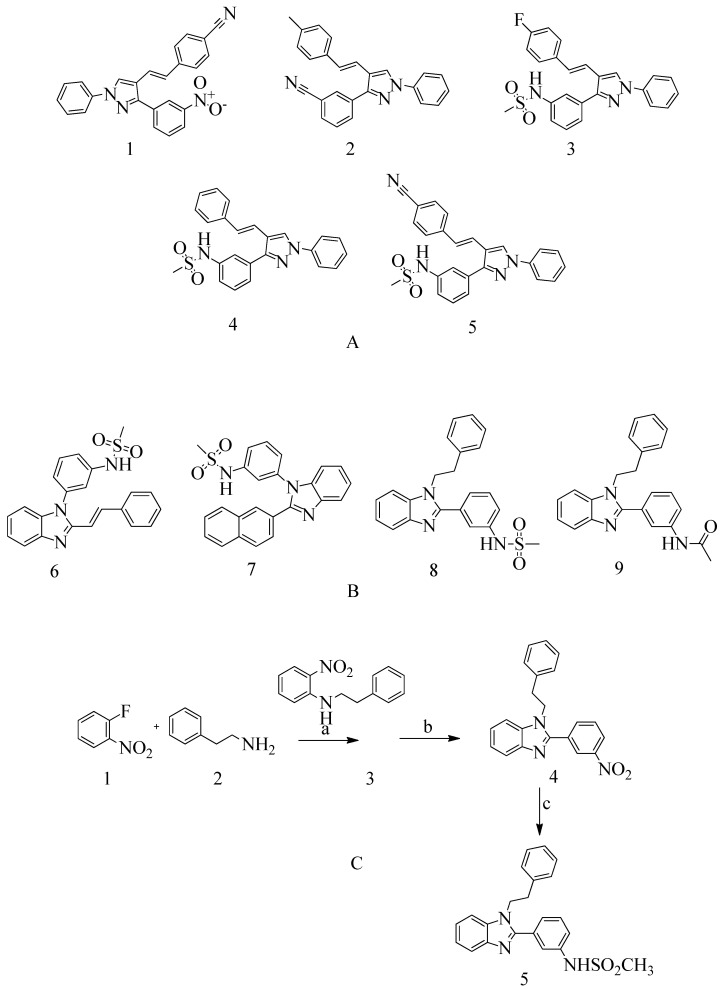
(**A**) Structures of novel pyrazole-based derivatives. (**B**) Structure of novel benzimidazole-based derivatives. (**C**) Synthesis route for benzimidazole-based derivates. Reaction details: Chemical agent names and reagents used are as follows: (**1**) 1-fluoro-2-nitrobenzene, added to (**2**) 2-phenylethanamine reagent, **a** K_2_CO_3_, DMF added and refluxed overnight, (**3**) 2-nitro-*N*-phenethylaniline obtained, to which **b** aldehyde, Sodium dithionite, and DMSO was added and kept overnight, which gives (**4**) 2-(3-nitrophenyl)-1-phenethyl-1*H*-benzo[*d*] imidazole, procured by adding reducing agent **c** SnCl_2_·2H_2_O, EtOAc, and final *N*-(3-(1-phenethyl-1*H*-benzo[*d*]imidazole-2yl)phenyl) methane sulfonamide was obtained [143,144].

**Figure 11 molecules-28-07008-f011:**
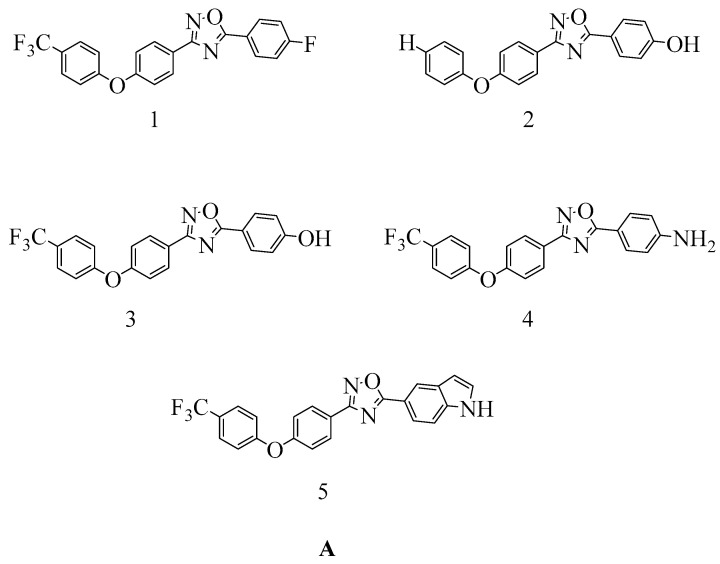
(**A**) Novel structure of 1,2,4-Oxadiazole-containing derivatives [150]. (**B**) Synthesis route for 1,2,4-oxadiazole-containing derivatives. Reagents used for reaction are as follows: (a) CuI, CsCO_3_, *N*,*N*-dimethylglycine HCl, and 1,4-dioxane, (b) Hydroxylamine ethanol, (c) Net(i-Pr)_2_ and CH_2_Cl_2,_ (d) THF and (Bu)_4_NF, (e) Toluene, (f) 1 eq. THF and TBAF, (g) NO_2_BzCl and toluene, and (h) Fe, H_2_O, HCl, and EtOH [147].

**Figure 12 molecules-28-07008-f012:**
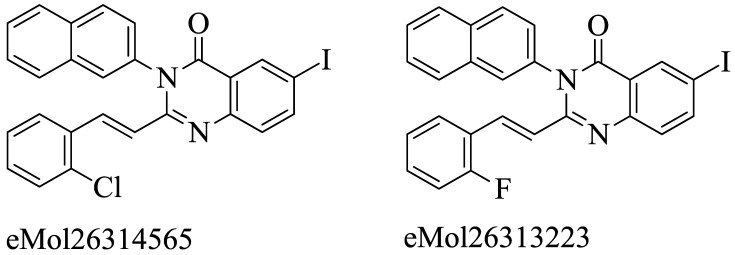
Non β-lactam allosteric inhibitors [151].

**Figure 13 molecules-28-07008-f013:**
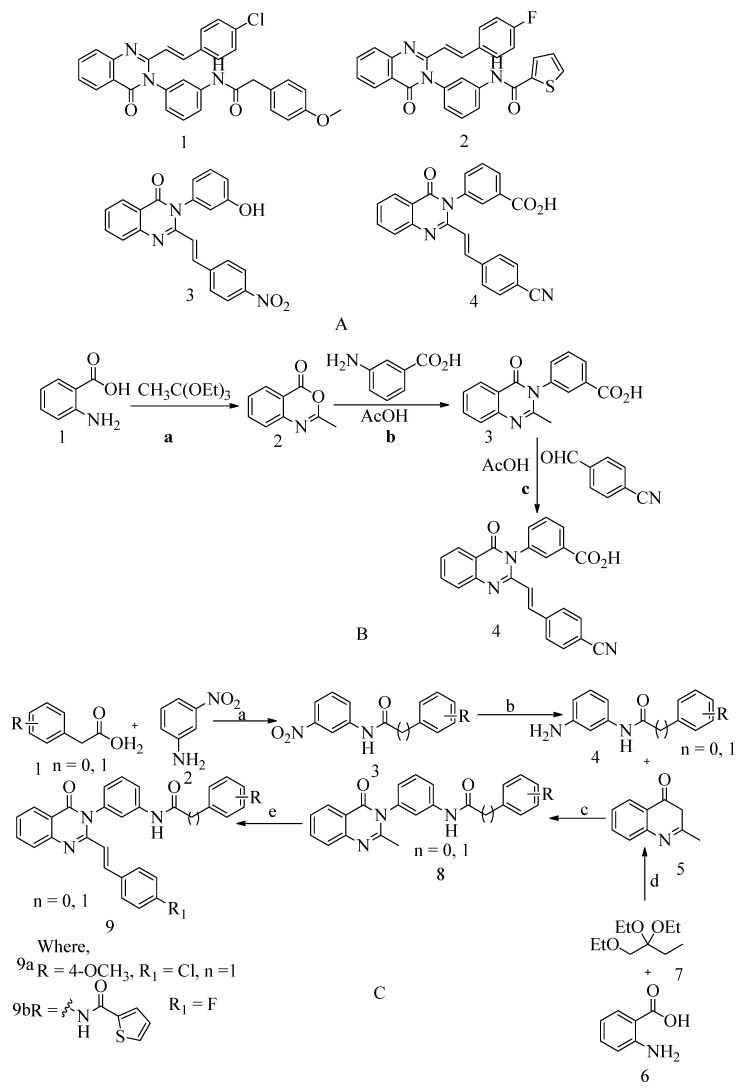
(**A**) Structure of 4-(3*H*)-quinazolinone. (**B**) Synthesis route of 4-(3*H*)-quinazolinone (compound **4** (see Figure 12A)); (a) quinazolinone and anthranilic acid, (b) acetic acid, (c) aldol-type condensation reaction and aromatic aldehyde [123,153]. (**C**) Synthesis route of 4-(3*H*)-quinazolinone (compounds **1** and **2** (see Figure 12A)). Reagents and conditions used for synthesis: (a) SOCl_2_, triethylamine (TEA), DCM, put overnight, (b) Fe, CaCl_2,_ Ethanol: H_2_O, refluxed for 5 to 6 h, (c) AcOH, refluxed for 1 to 2 h, (d) AcOH, refluxed for 1 to 2 h, and (e) AcOH, refluxed for 24 to 48 h.

**Figure 14 molecules-28-07008-f014:**
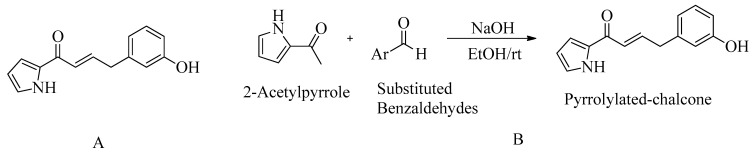
(**A**) Pyrrolylated-chalcone. (**B**) Synthetic route for pyrrolylated-chalcone. It is a Claisen-Schmidt condensation reaction. 2-Acetopyrrole reacting with substituted benzaldehyde in ethanolic sodium hydroxide for 24 h at room temperature [154].

**Figure 16 molecules-28-07008-f016:**
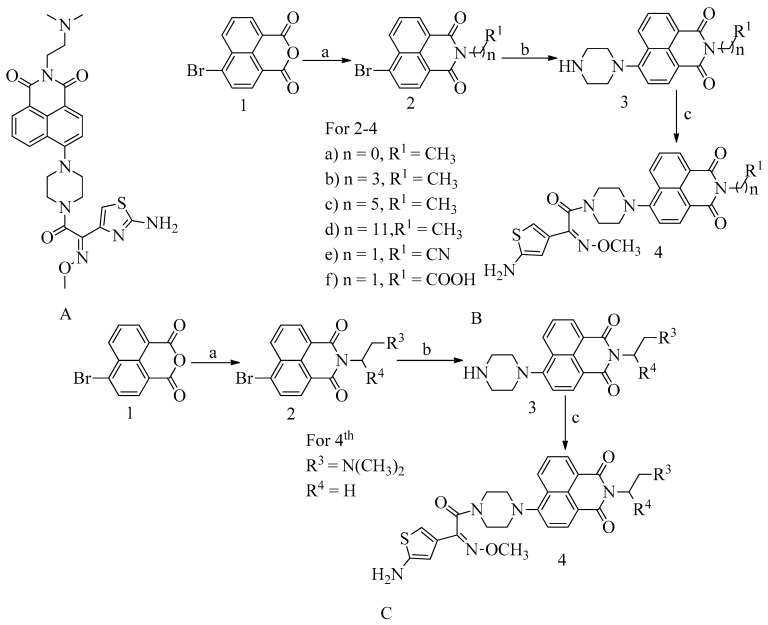
(**A**) A novel naphthalimide corbelled aminothiazoximes modulating PBP2a. (**B**) Preliminary procedure for naphthalimide corbelled aminothiazoximes. (**C**) Synthesis route for novel naphthalimide corbelled aminothiazoximes PBP2a modulator: (a) 2-(2-amino-4-thiazolyl)-2-methoxyiminoacetic (361 mg, 1.03 mmol), (b) 10 mL of ethanol, (c) 104 mg, 1.03 mmol of triethylamine at 25 C. 4a was used as starting material for the synthesis process. This contained an ethyl unit connected to the hydrophilic group. However, by the incorporation of the dimethylenediamine group, the MIC value was reduced to 0.05 µg/mL, effectively suppressing MRSA development to twice the amount of tetracycline, four times of vancomycin, eight times of ciprofloxacin, and thirty-two times of norfloxacin, from reported data [157].

**Figure 18 molecules-28-07008-f018:**
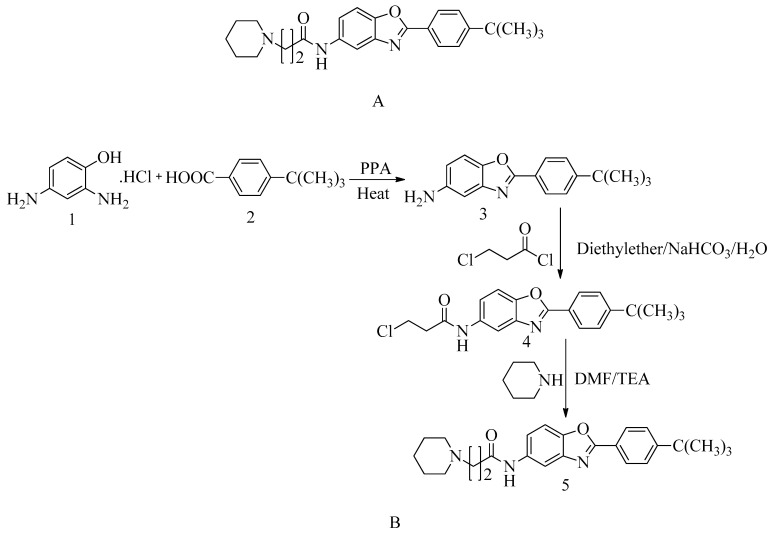
(**A**) Novel Benzoxazole derivatives. (**B**) Synthetic route for novel Benzoxazole derivatives [164].

**Figure 19 molecules-28-07008-f019:**
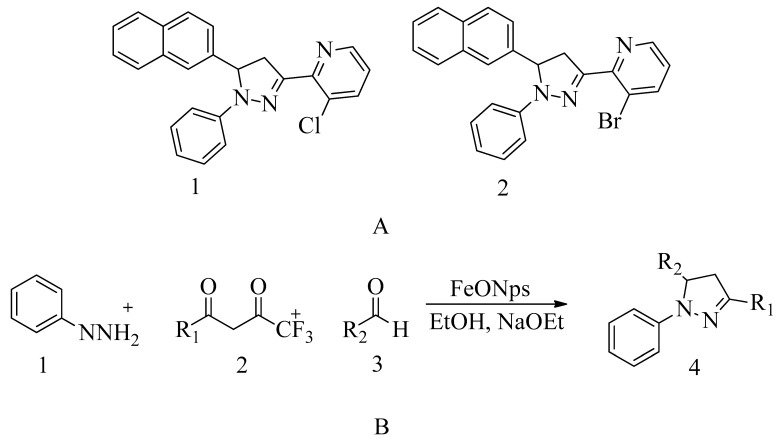
(**A**) Pyrazolylpyridine analogues. (**B**) Synthetic route for pyrazolylpyridine analogues; (1) phenylhydrazine, (2) 1,3 dicetone, and (3) pyridine carboxaldehyde, refluxed for 3–4 h using a basic medium (sodium ethoxide) [165].

**Figure 21 molecules-28-07008-f021:**
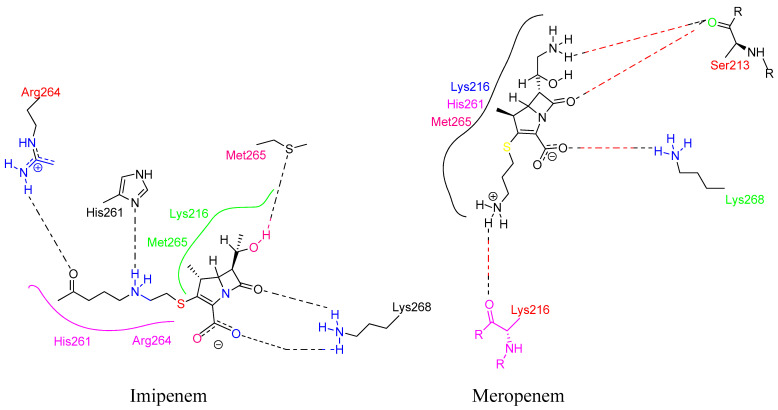
Binding pattern of imipenem and meropenem from 1VQQ crystal structure [178].

**Figure 22 molecules-28-07008-f022:**
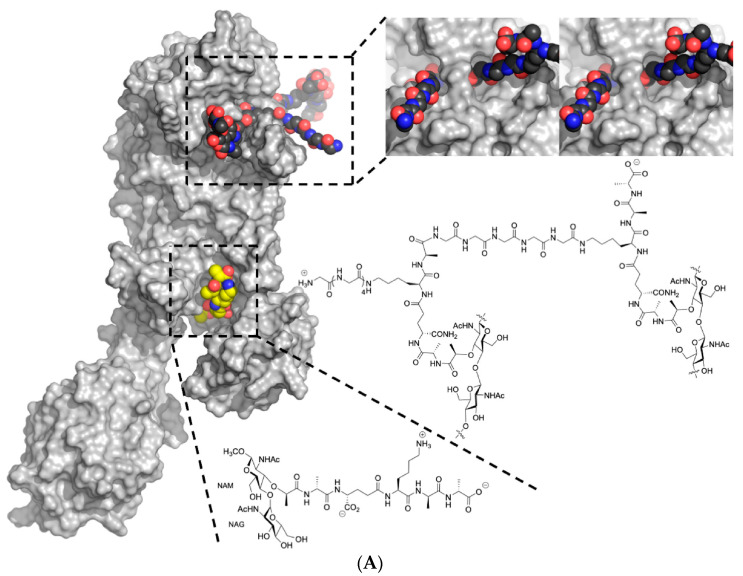
(**A**) The catalyst is enabled at the active site by binding peptidoglycan to the allosteric site [177] (Copyright permission @ACS). (**B**) PBP2a ligands detected by computer search. X-ray composition of *S. aureus* PBP2a is shown as a grey surface that the solvent can penetrate. At 6 o’clock, the protein attaches to the plasma membrane’s surface. Oxadiazole 1 is shown as a CPK model in a docked position at the active site at 1 o’clock, coded by atomic color type (carbon in dark gray, nitrogen in blue, oxygen in red, fluorine in green). After the enzyme rotates 90 degrees in the viewer’s direction along the Y-axis, the extension provides a stereo picture of the attached oxadiazole; all data are reported [177] (Copyright permission @ACS). (**C**) Chemicals most closely related to PBP2a. Interactions involving PBP2a and even berberine molecules are summarized in A1 and A2. The composition of PBP2a is shown in purple, while the composition of the berbamine compound is shown in red. The H bond between PBP2a and the berbamine molecule appears in A3. (Readers are directed to the web version of this article to clarify the colour references in the narrative of this figure: https://www.sciencedirect.com/science/article/pii/S235291482100304X?via%3Dihub#fig2, accessed on 4 October 2023) [176] (Copyright permission @Elsevier).

**Figure 23 molecules-28-07008-f023:**
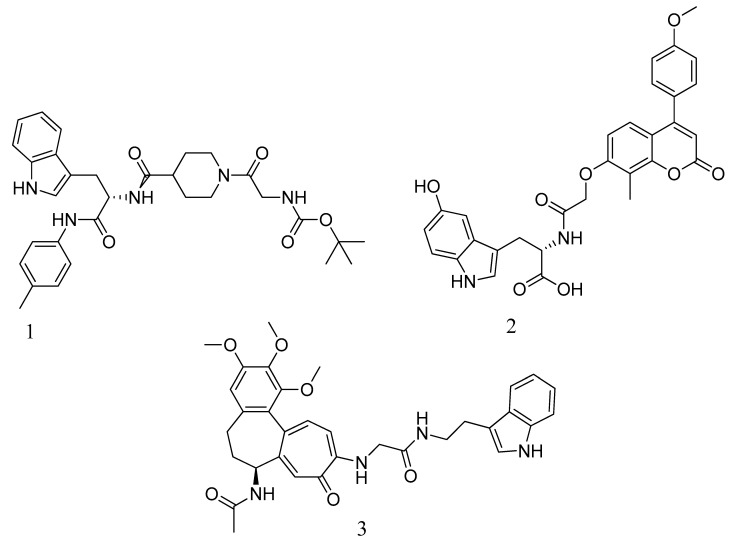
Virtual screening and biological evaluation of some of the novel PBP2a inhibitors.

**Figure 24 molecules-28-07008-f024:**
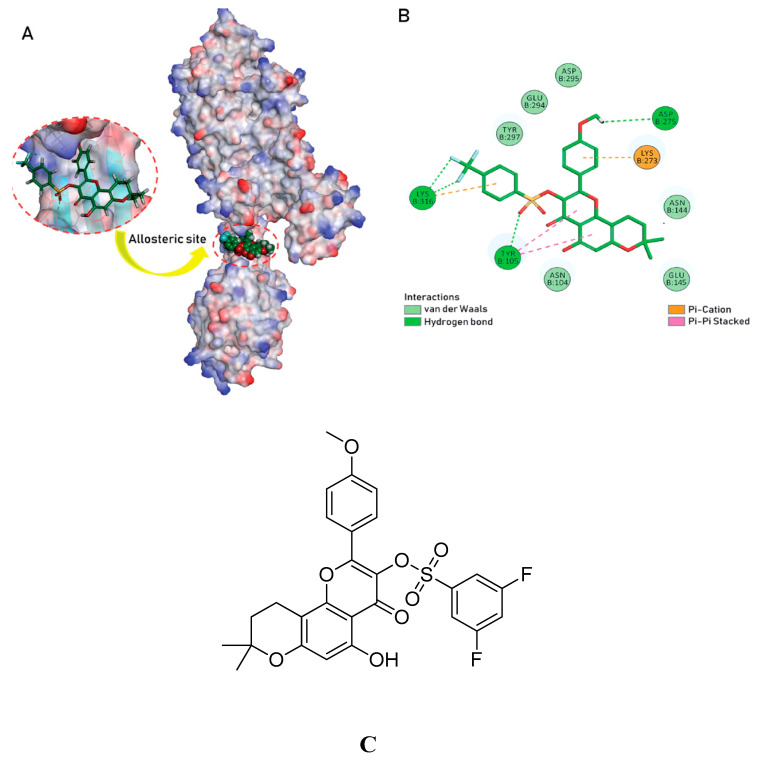
(**A**) 3D model of the interaction between compound with PBP2a (PDB 4CJN). (**B**) 2D binding mode within the receptor PBP2a’s allosteric site pocket. These figures were produced using the Discovery Studio 2017 R2 software (BIOVIA Discovery Studio 2017 R2 Client version 17.2) [180] (Copyright permission @Avoxa), (**C**) Structure of icariin derivative which was obtained as an active molecule from the virtual screening exercise [180].

**Figure 25 molecules-28-07008-f025:**
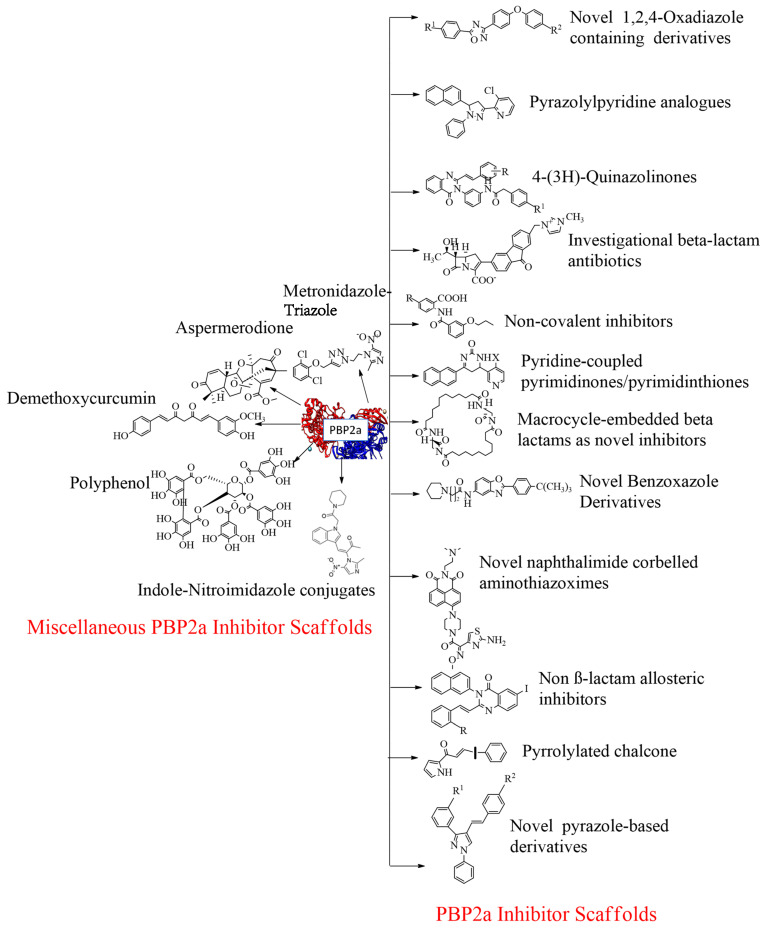
Schematic representation showing a target and different scaffold targeting PBP2a, especially the five membered and six membered heterocyclic scaffolds, including miscellaneous drugs.

**Table 3 molecules-28-07008-t003:** Crystal structure of PBP2a, their PDB ID, ligand bound to protein structure coupled with their date of publication in the Protein Data Bank (PDB) arranged in chronological sequence, and crystal structure resolutions and species of *Staphylococcus aureus*, with references.

Sr. No.	Release Year	PDB ID	Resolution	Bound Ligand	Species	Ref.
1	2019	6Q9N	2.5	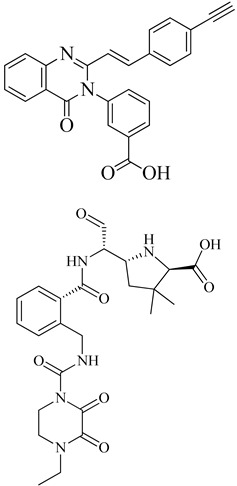	*S. aureus* subsp. Mu50	[121]
2	2019	6H5O	2.82	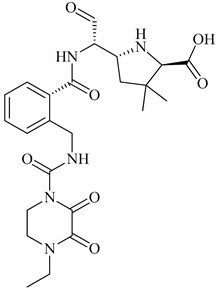	*S. aureus* Subsp. Mu50	[122]
3	2015	4CJN	1.947	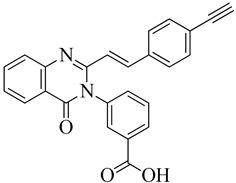	*S. aureus*	[123]
4	2013	3ZG0	2.6	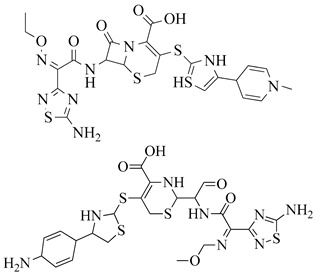	*S. aureus* Subsp. Mu50	[124]
5	2013	3ZFZ	2.25	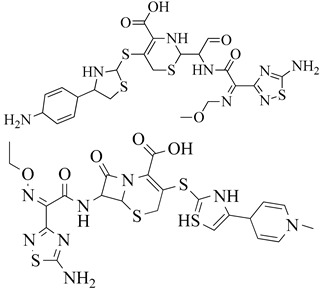	*S. aureus* subsp. Mu50	[124]
6	2002	1MWT	2.45	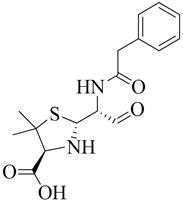	*S. aureus*	[115]
7	2002	1MWU	2.60	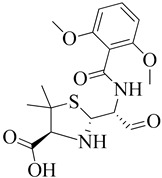	*S. aureus*	[115]

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
