# Peer review of "A Review on Five and Six-Membered Heterocyclic Compounds Targeting the Penicillin-Binding Protein 2 (PBP2A) of Methicillin-Resistant Staphylococcus aureus (MRSA)"

_molecules, 2023, doi:10.3390/molecules28207008_

Round 1

Reviewer 1 Report

Suggestions for Improvement:

Provide more detailed explanations of the mechanisms of action and modes of inhibition of PBP2A inhibitors. This will enhance the readers' understanding of the topic.

Include more recent studies and findings on PBP2A inhibitors to ensure the article reflects the latest research in the field.

Improve the organization and structure of the article by using subheadings to clearly delineate different sections and topics.

Consider adding visual aids such as figures or tables to illustrate key concepts or data, which can enhance the presentation and engagement of readers.

Plagiarism: After reviewing your article with iThenticate, we discovered instances of direct plagiarism and unattributed use of ideas from other sources. To enhance the quality of your work, we recommend inserting relevant citations and incorporating additional references to support the information you've presented.

The English language used in the article is appropriate and well-written, with no major grammatical or syntactical errors. However, a few minor improvements can be made to enhance clarity and readability.

Author Response

To,

The Reviewer

Molecules, MDPI

Dear Sir,

First of all, I thank you for your inputs, suggestions and comments on this manuscript, I highly appreciate your patience and time spent on our manuscript draft.

As per your suggestions, we have sorted out all the issues raised during the revision process, these are explained point wise as follows,

Query 1. Provide more detailed explanations of the mechanisms of action and modes of inhibition of PBP2A inhibitors. This will enhance the readers' understanding of the topic.

Response. Yes, we agree with your opinion and hence we have added more explanation regarding the mechanism and mode of action for the PBP2A inhibitors as well as development of resistance and its mechanism. We expect this will help readers better understand the topic.  

Query 2. Include more recent studies and findings on PBP2A inhibitors to ensure the article reflects the latest research in the field.

Response. Yes, we have updated the manuscript with about 8 new articles on PBP2A and its inhibitors published during 2022-23, now included in the revised form of the manuscript.

Query 3. Improve the organization and structure of the article by using subheadings to clearly delineate different sections and topics.

Response. We have improved the structure of the article by incorporating subheadings and subsections where feasible.

Query 4. Consider adding visual aids such as figures or tables to illustrate key concepts or data, which can enhance the presentation and engagement of readers.

Response. In the manuscript draft, we have tried to put most of the important aspects like the mechanisms, mode of action, synthetic routes for inhibitors, modelling study details in from of charts, diagrams, figures and chemical scaffolds for the better presentation and enjoyment of the readers.

Query 5. Plagiarism: After reviewing your article with iThenticate, we discovered instances of direct plagiarism and unattributed use of ideas from other sources. To enhance the quality of your work, we recommend inserting relevant citations and incorporating additional references to support the information you've presented.

Response. We have corrected the issues where instances of high similarity were observed, we thank the reviewer for pointing out areas which we missed in our checks. Now, we have corrected these and all corrections are highlighted for easy reference. We have inserted relevant references and has taken copyright permissions where necessary; all the information is updated in the manuscript and highlighted for quick reference.

Thank you,

Kind Regards

Reviewer 2 Report

1. Add a reference to Figure 2, cell wall assembly, or copyright permission from the original source.

2. Figures 8 (B) and (C) need to be redrawn in the chem sketch; they seem to be copied from the original source.

3. Figure 11 (B) needs to be redrawn in the chem sketch; it seems to be copied from the original source.

4. The chemical structures in Figure 22 are not of good quality.

5. Figure 24 does not have a good resolution. 3D and 2D models are not clear.

6. There are many English grammatical errors in the whole manuscript.

7. Overall the manuscript can be accepted after major revision.

Author Response

To,

The Reviewer

Molecules, MDPI

Dear Sir,

First of all, I thank you for your inputs, suggestions and comments on this manuscript, I highly appreciate your patience and time spent on our manuscript draft. We find your comments very helpful in improving the overall quality of the manuscript.

As per your suggestions, we have sorted out all the issues raised during the revision process, these are explained point wise as follows,

  1. Add a reference to Figure 2, cell wall assembly, or copyright permission from the original source.

Response. Yes, we have added reference to the figure 2, this was drawn by us with some information for refence articles. It is an original idea based on the referenced ideas.

  1. Figures 8 (B) and (C) need to be redrawn in the chem sketch; they seem to be copied from the original source.

Response. Yes, we have redrawn the figure 8 (B) and (C) with ChemDraw, these are updated in the draft and highlighted for easy reference.

  1. Figure 11 (B) needs to be redrawn in the chem sketch; it seems to be copied from the original source.

Response. Yes, we have redrawn the figure 11 (B) with ChemDraw, these are updated in the draft and highlighted for easy reference.

  1. The chemical structures in Figure 22 are not of good quality.

Response. We have improved the figure 22, now the images are replaced with the original ones from the reference article, copyright permission has been obtained.

  1. Figure 24 does not have a good resolution. 3D and 2D models are not clear.

Response. We have improved the figure 24, now the model images are replaced with the original ones from the reference article, copyright permission has been obtained.

  1. There are many English grammatical errors in the whole manuscript.

Response. We have gone through the article draft and has corrected the grammatical errors from the manuscript.

  1. Overall, the manuscript can be accepted after major revision

Response. We have corrected each of the suggested revisions.

Thank you,

Kind regards

Round 2

Reviewer 1 Report

-The suggestion to provide more detailed explanations of the mechanisms of action and modes of inhibition of PBP2A inhibitors has been comprehensively addressed. The authors expanded their explanations to offer readers a comprehensive understanding of the topic. The additional content delves into the intricacies of both the mechanism of action and the modes of inhibition, including insights into the development of resistance and its underlying mechanisms. These improvements have significantly contributed to the clarity of the manuscript.

-In response to the recommendation to include more recent studies and findings on PBP2A inhibitors, the authors have diligently updated the manuscript. They incorporated information from eight new articles published between 2022 and 2023, ensuring that their work reflects the latest research in the field. This additional content strengthens the overall relevance and currency of the manuscript.

-The suggestion to improve the organization and structure of the article through the use of subheadings has been implemented. The authors have restructured the manuscript, incorporating subheadings and subsections where appropriate. This adjustment enhances the overall readability and clarity, facilitating a smoother flow of information for the readers.

-Regarding the inclusion of visual aids such as figures or tables, the authors have integrated various visual elements into the manuscript. Charts, diagrams, figures, and chemical scaffolds now complement the text, providing a visual representation of key concepts, synthetic routes for inhibitors, and modeling study details. These additions have improved the presentation and engagement of readers.

-Concerning the issue of plagiarism highlighted through iThenticate, it is noted that the authors have meticulously addressed and corrected instances of direct plagiarism and unattributed use of ideas from other sources. Additionally, they have inserted relevant citations and incorporated additional references to support the information presented in the manuscript. All corrections have been meticulously highlighted for easy reference, ensuring transparency in their revisions.

Given the comprehensive nature of these revisions, I confidently reiterate that the manuscript now meets the high standards of Molecules. Therefore, I recommend that the article be considered for publication in one of its upcoming issues.

Minor editing of English language required

Author Response

To,

The Reviewer

Molecules, MDPI

Dear Sir,

We thank you for accepting our corrections. Also, we highly appreciate your time, suggestions and support during this process. 

Thank you,

Kind regards

Reviewer 2 Report

The manuscript has been revised according to the comments of the reviewer.

All the comments have been addressed and replies to reviewers' comments have been provided.

The revised manuscript still needs to be modified before final acceptance.

Following the comments:

1. Figure 8 (A) is not clear. All the structures need to be drawn again in the chem sketch.

2. Figure 11 (A), redraw the structures in the Chem sketch.

3. Figure 20, Structure (J) seems to be copied.  Redraw the structures in the Chem sketch.

Author Response

To,

The Reviewer

Molecules, MDPI

Dear Sir,

We thank you for your inputs, suggestions and comments on this manuscript in the second round of review process. We value your suggestions and has addressed them in the updated manuscript draft, all the changes are highlighted for your quick perusal. The pointwise corrections are as follows,

  1. Figure 8 (A) is not clear. All the structures need to be drawn again in the chem sketch.

Response: Yes, we have redrawn the structures and using chemDraw and the figure 8 (A) is now improved.

  1. Figure 11 (A), redraw the structures in the Chem sketch.

Response: Yes, we have redrawn the structures and using chemDraw and the figure 11 (A) is now improved.

  1. Figure 20, Structure (J)seems to be copied.  Redraw the structures in the Chem sketch.

Response: Yes, we have redrawn the structures and using chemDraw and the figure 20 (J) is now improved.

Thank you,

Best regards